

# MAX-DOAS measurements of HONO slant column densities during the MAD-CAT Campaign: inter-comparison and sensitivity studies on spectral analysis settings

Yang Wang[1,2], Steffen Beirle[1], Francois Hendrick[3], Andreas Hilboll[4,5], Junli Jin[6,7], Aleksandra A. Kyuberis[8], Johannes Lampel[9,1], Ang Li[2], Yuhan Luo[2], Lorenzo Lodi[10], Jianzhong Ma[6], Monica Navarro[11], Ivan Ortega[12], Enno Peters[4], Oleg L. Polyansky[10,8], Julia Remmers[1], Andreas Richter[4], Olga Puentedura Rodriguez[11], Michel Van Roozendael[3], André Seyler[4], Jonathan Tennyson[10], Rainer Volkamer[12], Pinhua Xie[2,13], Nikolai F. Zobov[8] and Thomas Wagner[1]

[1] Max Planck Institute for Chemistry, Mainz, Germany
[2] Anhui Institute of Optics and Fine Mechanics, Key laboratory of Environmental Optics and Technology, Chinese Academy of Sciences, Hefei, 230031, China
[3] Belgian Institute for Space Aeronomy – BIRA-IASB, Brussels, Belgium
[4] Institute of Environmental Physics, University of Bremen, Bremen, Germany
[5] Center for Marine Environmental Sciences (MARUM), University of Bremen, Bremen, Germany
[6] Chinese Academy of Meteorological Sciences, Beijing, China
[7] CMA Meteorological Observation Centre, Beijing, China
[8] Institute of Applied Physics, Russian Academy of Sciences, Nizhny Novgorod, Russia
[9] Institute of Environmental Physics, University of Heidelberg, Heidelberg, Germany
[10] Department of Physics and Astronomy, University College London, Gower St, London WC1E 6BT, UK
[11] Area de Investigación e Instrumentación Atmosférica, INTA, Torrejón de Ardoz, Spain
[12] Department of Chemistry and Biochemistry, University of Colorado, Boulder, CO, USA
[13] CAS Center for Excellence in Urban Atmospheric Environment, Institute of Urban Environment, Chinese Academy of Sciences, Xiamen, 361021, China

*Correspondence to*: Yang Wang (y.wang@mpic.de); Ang Li (angli@aiofm.ac.cn)

**Abstract.**

In order to promote the development of the passive DOAS technique the Multi Axis DOAS – Comparison campaign for Aerosols and Trace gases (MAD-CAT) was held at the Max Planck Institute for Chemistry in Mainz, Germany from June to October 2013. Here, we systematically compare the differential slant column densities (dSCDs) of nitrous acid (HONO) derived from measurements of seven different instruments. We also compare the tropospheric difference of SCDs (delta SCD) of HONO, namely the difference of the SCDs for the non-zenith observations and the zenith observation of the same elevation sequence. Different research groups analysed the spectra from their own instruments using their individual fit software. All the fit errors of HONO dSCDs from the instruments with cooled large-size detectors are mostly in the range of 0.1 to $0.3 \times 10^{15}$ molecules cm$^{-2}$ for an integration time of 1 min. The fit error for the mini MAX-DOAS is around $0.7 \times 10^{15}$ molecules cm$^{-2}$. Although the HONO delta SCDs are normally smaller than $6 \times 10^{15}$ molecules cm$^{-2}$, consistent time series of HONO delta SCDs are retrieved from the measurements of different instruments. Both fits with a sequential Fraunhofer reference spectrum (FRS) and a daily noon FRS lead to similar consistency. Apart from the mini-MAX-DOAS, the



systematic absolute differences of HONO delta SCDs between the instruments are smaller than $0.63\times10^{15}$ molecules cm$^{-2}$. The correlation coefficients are higher than 0.7 and the slopes of linear regressions deviate from unity by less than 16% for the elevation angle of 1°. The correlations decrease with an increase of elevation angle. All the participants also analysed synthetic spectra using the same baseline DOAS settings to evaluate the systematic errors of HONO results from their

respective fit programs. In general the errors are smaller than $0.3\times10^{15}$ molecules cm$^{-2}$, which is about half of the systematic difference between the real measurements.

The differences of HONO delta SCDs between retrieved in the selected three spectral ranges 335-361nm, 335-373nm and 335-390nm are considerable (up to $0.57\times10^{15}$ molecules cm$^{-2}$) for both real measurements and synthetic spectra. We performed sensitivity studies to quantify the dominant systematic error sources and to find a recommended DOAS setting in

the three spectral ranges. The results show that water vapour absorption, temperature and wavelength dependence of $O_4$ absorption, temperature dependence of Ring spectrum, and polynomial and intensity offset correction all together dominate the systematic errors. We recommend a fit range of 335-373nm for HONO retrievals. In such fit range the total systematic uncertainty from different sources is about $0.87\times10^{15}$ molecules cm$^{-2}$, much smaller than those in the other two ranges. Meanwhile the systematic bias of the fitted from the simulated real HONO delta SCDs is also smallest in 335-373nm (about

$0.02\times10^{15}$ molecules cm$^{-2}$). The typical random uncertainty is estimated to be about $0.16\times10^{15}$ molecules cm$^{-2}$, which is only 25% of the total systematic uncertainty for most of the instruments in the MAD-CAT campaign. In summary most of the MAX-DOAS instruments can well observe the signals of atmospheric HONO absorption in case of HONO delta SCDs higher than $0.2\times10^{15}$ molecules cm$^{-2}$. However, systematic uncertainties limit the reliability of the results.

## 1 Introduction

Nitrous acid (HONO) is an important precursor of the OH radical, which prominently controls the self-cleaning capacity of the tropospheric atmosphere (Alicke et al., 2003; Kleffmann et al., 2005; Acker et al., 2006; Monks et al., 2009; Elshorbany et al., 2010). The gas-phase reaction of NO with the OH radical (Stuhl and Niki, 1972 and Pagsberg et al., 1997) mostly determines the daytime HONO concentration. However, recent field measurements (Neftel et al., 1996; Kleffmann et al., 2005; Sörgel et al., 2011; Li et al., 2012 and 2014 and Wong et al., 2012) and laboratory studies (Akimoto et al., 1987;

Rohrer et al., 2005) reported much larger HONO concentrations than predicted by the gas-phase reactions. These findings imply some missing daytime sources of HONO. Laboratory and field studies suggest that the missing daytime sources consist of heterogeneous reactions on various surfaces such as the ground, forests, buildings, and aerosols (Su et al., 2008 and 2011; Li et al., 2014; and references therein), emissions from soil (Su et al., 2011 and references therein), and a potential gas-phase reaction between $HO_x$ and $NO_x$ (Li et al., 2014).

The overall effect of the proposed missing HONO sources in the troposphere remains widely unknown because of the lack of measurements of HONO and its relevant precursor species at higher altitudes above the ground (Li et al., 2014). The surface HONO concentrations can be well quantified by ground-based in-situ instruments, like the LOPAP (long path absorption



photometer) technique (Heland et al., 2001; Kleffmann et al., 2006 and Li et al., 2012) and long-path DOAS (Trick, 2004 and references therein). Besides these techniques, four other optical absorption techniques have been used for the detection of HONO, i.e., cavity ring down spectroscopy (Wang and Zhang, 2000), FTIR spectroscopy (Hanst et al., 1982), tunable diode laser spectroscopy (Schiller et al., 2001) and CE-DOAS (Hoch et al., 2014). To quantify the distribution of HONO in

elevated layers of the troposphere, the in-situ LOPAP technique has been mounted aboard on an airship Zeppelin platform (Li et al., 2014). However, because of the large cost of operating such a flight platform, the corresponding data sets are limited in time and space.

Since about 15 years ago, the Multi Axis - Differential Optical Absorption Spectroscopy (MAX-DOAS) technique, which is based on the DOAS spectral analysis technique (Platt and Stutz, 2008, and references therein),  has been widely used owing

to its potential to retrieve the vertical distribution of trace gases and aerosols in the lower part of the troposphere from scattered sunlight spectra recorded at multiple elevation angles using relatively simple and low-cost ground-based instrumentation (Hönninger and Platt, 2002; Bobrowski et al., 2003; Van Roozendael et al., 2003; Hönninger et al., 2004; Wagner et al., 2004 and Wittrock et al., 2004). Hendrick et al. (2014) reported the first MAX-DOAS measurements of vertical column densities (VCDs) and near-surface volume mixing ratios (VMRs) of HONO in the Beijing area, China.

Because of its simple and automatic operation at the ground, MAX-DOAS is well suited to continuously acquire HONO vertical distributions over longer time periods. However, due to the typically low HONO VMRs in the troposphere (between about 50 ppt to 2000 ppt near the surface in urban areas (Li et al., 2012)) and the moderate cross section with the maximum of about $5 \times 10^{-19}$ cm$^2$ molecules$^{-1}$ in the UV range, the atmospheric HONO absorption is rather weak, and it can also be systematically interfered by strong absorptions of other trace gases (e.g. $O_3$ and $NO_2$) and instrument-related spectral

structures. So far few efforts have been devoted to study these error sources in HONO DOAS fits of MAX-DOAS spectra. Furthermore, many research groups have developed their own MAX-DOAS instruments equipped with various types of spectrometers, detectors, and entrance optics. Thus the inter-comparison of HONO measurements and retrieval results from different MAX-DOAS instruments is essential to evaluate MAX-DOAS HONO results and associated uncertainties.

The Multi Axis DOAS – Comparison campaign for Aerosols and Trace gases (MAD-CAT) held at the Max Planck Institute

for Chemistry in Mainz, Germany in June and July 2013 (http://joseba.mpch-mainz.mpg.de/mad_cat.htm). During MAD-CAT campaign,  eleven MAX-DOAS instruments from all of the eleven groups were operated in parallel, providing an opportunity to assess the consistency of different HONO measuring MAX-DOAS systems for the first time. In this study, only the direct output values, namely the slant column density (SCD) of HONO in the troposphere, derived from the spectral analysis (DOAS fit) of the acquired MAX-DOAS spectra are compared between the instruments and discussed with respect

to their systematic error sources based on sensitivity tests. The inter-comparison activities in this study follow similar work done for $NO_2$ and HCHO during the Cabauw Intercomparison campaign of Nitrogen Dioxide measuring Instruments (CINDI) (Piters et al., 2012) in The Netherlands in June–July 2009 (Roscoe et al., 2010 and Pinardi et al., 2013).

In addition to the measured spectra, a set of synthetic spectra generated by the SCIATRAN radiative transfer model (RTM) (Rozanov et al., 2014) was analysed for the first time. These spectra are simulated based on various atmospheric scenarios



including not only HONO, but also other relevant trace gases and aerosols. Because the HONO SCDs of the synthetic spectra are known, the bias of the retrieved SCDs from the true values can be easily quantified.

This paper is structured as follows. Section 2 gives an overview of the MAD-CAT campaign and participating instruments. Section 3 presents inter-comparison results of the HONO SCD derived from real measurements and synthetic spectra between the participants. In Section 4 we focus on sensitivity tests to assess possible interferences in the HONO SCD retrievals. Recommended analysis settings are given together with an error budget in Section 5. The conclusions are presented in Section 6.

## 2 Field experiment

### 2.1 The MAD-CAT inter-comparison campaign

The Multi Axis DOAS – Comparison campaign for Aerosols and Trace gases (MAD-CAT) was held at the Max Planck Institute for Chemistry (MPIC) in Mainz, Germany. The measurement site is located in the outskirts of the city of Mainz, close to agricultural fields in the west. The large city of Frankfurt am Main with about 0.7 million inhabitants is about 30 km away from the measurement site in the northeast. All the MAX-DOAS instruments from eleven participating groups were operated on the roof of MPIC during the intensive measurement phase from 7 June until 6 July, 2013. Only the measurements in the period from 12 June to 5 July are included in the HONO inter-comparison activity due to the time coverage of the participating instruments. Although many of the instruments are designed to measure in various azimuth angles, in this study only the measurements in the main azimuth direction of 51 °northeast are included. The same elevation angle sequence of 1 °, 2 °, 3 °, 4 °, 5 °, 6 °, 8 °, 10 °, 15 °, 30 °and 90 °was applied by all instruments. A description of the MAD-CAT measurement campaign can be found at http://joseba.mpch-mainz.mpg.de/mad_cat.htm. Data from this campaign has been so far used e.g. in Ortega et al. (2015), Lampel et al. (2015 and 2016b) and Peters et al. (2016).

### 2.2 Instruments

Seven of all the eleven groups participated in the HONO inter-comparison activity. The primary information on the instruments is listed in Tab. 1. The instruments use different types of detectors, spectrometers and optical systems. Except the instrument of CMA, which is a "mini-MAX-DOAS" from Hoffmann Messtechnik GmbH in Germany, all other instruments are self-built. Only the mini-MAX-DOAS instrument integrates the entrance optics and fibre coupled spectrograph in a hermetically sealed metal box positioned outdoors. The other six instruments have two separate parts: one is indoors with a fibre coupled spectrograph located in a temperature stabilised box; the other is outdoors with the entrance optics and pointing telescope. The Heidelberg and CMA instruments used small-size compact spectrographs. The other instruments were equipped with large size spectrometers with thermoelectrically cooled imaging CCD array detectors. In this study, measurements during the period from 12 June to 5 July are considered, which is covered by most of the instruments.



The data availability from the different instruments is shown in Supplementary Fig. S1. Because the different instruments applied different integration times (see Tab. 1) and scanning speeds, and also partly performed measurements in other directions, different numbers of elevation sequences per hour are acquired (see Fig. S1 in the Supplement). Typical numbers of elevation sequences per hour range from 2.8 (BIRA) to 9.4 (MPIC).

## 3. Results and inter-comparison

### 3.1 Baseline HONO analysis settings

HONO presents prominent absorption structures in the spectral range from 335 to 390 nm. The DOAS technique (Platt and Stutz, 2008, and references therein) can be applied to spectra of scattered sunlight to retrieve SCDs of HONO. The baseline DOAS fit setting for the inter-comparison activity was determined based on the experiences in Hendrick et al. (2014) and is described in Table 2. Instead of the spectral range from 337 to 375 nm used in Hendrick et al. (2014), a slightly different range (335 to 373 nm) is used here because of the limitation of the upper edge of the wavelength range of the Bremen instrument. Absorption cross sections of HONO, $NO_2$, $O_3$, BrO, $O_4$, and HCHO were convolved to the spectral resolution of the individual instruments and included in the fit. The solar $I_0$ correction was applied to the $O_3$ and $NO_2$ cross sections (Aliwell et al., 2002). To correct the wavelength dependence of the $NO_2$ AMF (see Section 4.5), the Taylor series terms of $\lambda\sigma_{NO_2}$ and $\sigma_{NO_2}^2$ (with $\lambda$ the wavelength, and $\sigma_{NO_2}$ the $NO_2$ cross section) (Puķīte et al., 2010) were included in the fit. The effect of rotational Raman scattering was considered by including a Ring spectrum (Shefov 1959; Grainger and Ring, 1962; Chance and Spurr, 1997; Solomon et al., 1987; Wagner et al., 2009). The Ring spectrum was calculated according to Chance and Spurr (1997) based on the high resolution solar atlas of Kurucz et al. (1984) for a temperature of 250 K and convoluted to the respective instrumental resolution. To account for different wavelength dependencies of the filling-in in clear and cloudy skies, an additional Ring spectrum as described in Wagner et al. (2009) was included.

To correct for the strong Fraunhofer lines, a Fraunhofer reference spectrum (FRS) was included in the DOAS fit. Thus the SCDs from the DOAS fits actually represent the difference between the SCD of the measured spectra and the FRS. This difference is usually referred to as differential SCD (dSCD). From the dSCDs, so called "tropospheric difference" SCDs (delta SCDs) (Hönninger et al. 2004) can be extracted by two methods:

a) By using, as the FRS, the zenith measurement in the same elevation sequence as the off-axis spectra (Clémer et al., 2010; Peters et al., 2012; Hendrick, 2014). Such a FRS is referred to as "sequential FRS" in this study.

b) By first retrieving the dSCDs for all the elevation angles (including zenith view) using a single zenith spectrum (typically around noon) on one day. This FRS is referred to as "daily noon FRS" in this study. From the derived dSCDs the delta SCD is calculated by subtracting the dSCD of the zenith spectrum from the respective dSCD of the off-axis spectra in the corresponding elevation sequence (Hönninger et al. 2004; Pinardi et al., 2013 and Ma et al., 2013).

In principle the delta SCDs from the two schemes should be the same, but the fits using a daily noon FRS are usually more strongly affected by changes of instrumental properties and interferences of stratospheric absorptions (e.g. $O_3$) than those





using a sequential FRS. To quantify the effect of the different types of FRS, we compare the HONO delta SCDs from both methods.

## 3.2 Results of HONO delta SCDs and dSCDs, and fit errors

Figure 1 presents examples of DOAS fits of one spectrum measured by the AIOFM instrument using the baseline setting

with either the sequential FRS (left) or daily noon FRS (right), respectively. The fits were performed using the WINDOAS software (Fayt and van Roozendael, 2009). The HONO absorption structures are well retrieved using both types of FRS. However, the results for the trace gases, especially for those with stratospheric contributions, e.g. $O_3$, $NO_2$, and BrO, are systematically larger using a daily FRS, because around noon the stratospheric light paths are much shorter than during sunset or sunrise. Also the root mean square (RMS) of the fit residual of $4.5 \times 10^{-4}$ (corresponding to a HONO dSCD error of

$2.6 \times 10^{14}$ molecules cm$^{-2}$) using a sequential FRS is slightly smaller than the RMS of $5.7 \times 10^{-4}$ (corresponding to HONO dSCD error of $3.1 \times 10^{14}$ molecules cm$^{-2}$) using a daily noon FRS.

Figure 2a shows the hourly averaged HONO delta SCDs at 1 °elevation angle derived from the measurements of the AIOFM instrument during the whole comparison period; Figure 2b shows the corresponding averaged diurnal variation. A large variability of the HONO delta SCDs is found between -$1 \times 10^{15}$ molecules cm$^{-2}$ (negative value probably due to the effect of

15 water vapour absorption, see Section 4.1) and $5 \times 10^{15}$ molecules cm$^{-2}$. In general, the highest values are found in the morning. In addition to the HONO delta SCDs, delta SCDs of oxygen dimer ($O_4$) are also shown in Fig. 2. Since the atmospheric $O_4$ mixing ratio is rather constant and well known, variations of the $O_4$ delta SCDs can be used as an indicator for variations of the atmospheric absorption path length (e.g. Erle et al., 1995; Hönninger et al., 2004; Sinreich et al., 2013; Wang et al., 2014; and references therein). As can be seen in Fig. 2b, the delta SCDs of HONO and $O_4$ show systematically different diurnal

variations indicating that the observed variation of the HONO delta SCDs is not an artefact caused by the variation of the light path length but mainly reflects the variation of the atmospheric HONO concentration.

HONO dSCDs are retrieved by each group using the same baseline analysis settings as shown in Tab. 2. For the inter-comparison of the different data sets, we first averaged the HONO dSCDs for individual elevation angles of each instrument during periods of one hour, in which all the instruments have more than two measurement sequences (see Fig. S1 in the

25 Supplement).

Figure 3 shows an example of the time series of the hourly averaged HONO delta SCDs for individual elevation angles derived from each instrument using the fits with a sequential or a daily noon FRS as well as the HONO dSCDs using a daily noon FRS on 3 July 2013. The results for the five selected elevation angles are shown in Fig. 3 and those for other elevation angles are shown in Supplementary Fig. S2. On this day all the instruments provide credible data, and also rather large

HONO dSCDs and delta SCDs are observed in the morning, in particular at lower elevation angles. Note, however, that because of unknown instrumental problems, CMA and Boulder didn't participate in the comparisons of the delta SCDs for a sequential FRS and dSCDs for a daily noon FRS, respectively. As can be seen in Fig. 3, much better agreements between the





instruments are obtained for the delta SCDs than for the dSCDs; all the instruments capture well the diurnal evolution and elevation angle dependence of the HONO delta SCDs.

Figure 4a presents the hourly averaged fit errors of the HONO dSCDs using a daily noon FRS plotted against the solar zenith angle (SZA) for the whole comparison period. The fit errors depend on the random and systematic structures of the spectral residual. Systematic structures are mainly caused by instrumental shortcomings, possible non-considered atmospheric absorption structures, as well as imperfect corrections of rotational Raman scattering, temperature dependences of atmospheric absorptions, and wavelength dependences of absorption light paths (namely air mass factor (AMF)). Increasing fit errors with increasing SZA are found for all the instruments due to the reduction of the solar radiance and the increase of stratospheric absorptions (e.g. ozone). The largest fit error is found for the CMA instrument due to the relatively low signal to noise ratio of the detector. The second largest fit error is found for the MPIC instrument due to the very short integration time (Fig. 4b). The fit errors of other instruments are similar and in the range $0.15 \times 10^{15}$ molecules cm$^{-2}$ to $0.5 \times 10^{15}$ molecules cm$^{-2}$ for SZA <60 °. Because the random noise of an instrument depends on integration time, which is different for different instruments (see Fig. 4b), the fit errors are scaled to a typical integration time of 1 minute in order to make the results directly comparable (see Fig. 4c). Note that we applied a linear scaling, which is not strictly correct since the photon noise shows a square-root dependency of the number of observed photons. However, since the MAX-DOAS instruments are not radiometrically calibrated, we applied a linear scaling to achieve a first order normalisation for the effect of the integration time. Similar normalized fit errors are found for the instruments using cooled large-size detectors (BIRA, Bremen, AIOFM, Boulder and MPIC). Although both the Heidelberg and CMA instruments use compact spectrometers, Fig. 4c demonstrates that the Avantes spectrometer (http://www.avantes.com) in the Heidelberg instrument has a much lower noise level than the ocean optics USB 2000 (http://oceanoptics.com/) in the CMA instrument. Figure 4d also indicates that the fit errors with daily noon FRS are generally higher than those with sequential FRS for all instruments. The difference is especially large for the Bremen instrument, probably due to a known temperature stability problem of the spectrometer during the MADCAT campaign.

Because of the instrumental straylight, possible imperfect correction of the dark current and electronic offset signal in the measured spectra , and vibrational Raman scattering (Lampel et al., 2015), usually an intensity offset correction is included in the DOAS fit procedure (e.g. Noxon, 1975; Fayt and van Roozendael, 2009). However, the effect of spectral straylight and its correction by the intensity offset fit could interfere with retrievals of the species with low optical depths (Coburn, 2011). It is known that spectral straylight typically depends on the sky colour. Thus the strength of the corresponding spectral interferences also depends on the actual sky condition during the measurement. Supplementary Figs. S3a and b present the averaged diurnal variations of the intensity offset at 354 nm for individual instruments derived from the fits with a daily noon FRS and sequential FRS, respectively. The fitted intensity offsets for most of the instruments are lower than 0.01 for analyses with both types of FRS. Much larger offsets are found only for the CMA instrument, especially in the morning and afternoon.





The shift of the wavelength calibration of the measured spectra with respect to the FRS is mathematically determined and corrected in the DOAS fit procedure. The wavelength shift can be caused by the tilt effect (typically <2pm) (Lampel et al., 2016b) and dominated by the mechanical deformation of a spectrometer, which is usually sensitive to variations of the ambient temperature. The averaged diurnal variations of the shifts derived from the fits with a daily noon FRS and a sequential FRS are compared between the different instruments in Supplementary Fig. S3c and d, respectively. As expected, the shifts for the sequential FRS are much smaller than those for the daily noon FRS.

### 3.3 Statistical inter-comparisons

In this Section, we apply the statistical analysis method introduced in Roscoe et al. (2010) and Pinardi et al. (2013) to the inter-comparison of HONO results (delta SCDs and dSCDs) from the individual instruments. The results from the Heidelberg, BIRA, Bremen, and AIOFM instruments are averaged as reference values because of their almost full time coverage, low fitting errors, and good agreement. In the following discussions, we assume the reference values as the truth, but this is not necessarily the case. We implement two methods for the comparisons:

1) To derive an overview of the general agreement between the retrieval results by different instruments for the whole measurement period, mean absolute differences and standard deviations of HONO results from the reference values are summarized for individual elevation angles. Meanwhile a set of histograms of the absolute differences is prepared.

2) To investigate how well the different instruments capture the diurnal variation of the HONO dSCDs, for eight selected days with pronounced diurnal variations of the HONO dSCDs and delta SCDs (12, 15, 17, 18 and 30 June as well as 1, 2 and 3 July 2013), a set of scatter plots with linear regressions of the results from the different instruments against the reference values is prepared.

We performed the two comparisons for the two HONO delta SCDs from the fits with a sequential FRS or a daily noon FRS as well as for the HONO dSCDs with a daily noon FRS. Note that in the figures below, only the results for the elevation angles 1°, 3°, 5°, 8°, 15° (also 90° only for dSCDs with a daily noon FRS) are shown, but similar conclusions can be drawn for the other elevation angles. The mean absolute differences and standard deviations as well as the correlation coefficients, slopes and intercepts of the linear regressions derived from comparisons of HONO dSCDs (noon FRS) and delta SCDs of different instruments with respect to the reference values are presented in Fig. 5. In general linear correlations of the three HONO results decrease with an increase of elevation angle for all the instruments, probably due to the low values and small value ranges. However, there are no dependences of the absolute differences and standard deviations on the elevation angles for most of the instruments, except the CMA instrument. The comparison results of HONO delta SCDs derived the fits with either a sequential FRS or a daily noon FRS are quite similar with each other, however mostly different from those of HONO dSCDs with a daily noon FRS, for individual instruments. In the following we separately discuss the comparisons of three HONO results.

For the HONO delta SCDs with a sequential FRS, histograms of the absolute differences from the reference values for each instrument are shown in Supplementary Fig. S4. A symmetric and quasi-Gaussian shape of the histograms is found for all



the instruments, but differences of the Gaussian full width at half maximum (FWHM) (represented by standard deviation) and the mean absolute differences between the instruments are considerable. Larger standard deviations are found for MPIC ($\sim0.6\times10^{15}$ molecules cm$^{-2}$), Boulder ($\sim0.3\times10^{15}$ molecules cm$^{-2}$), and Bremen ($\sim0.3\times10^{15}$ molecules cm$^{-2}$) compared to the other instruments ($\sim0.16$ to $\sim0.28\times10^{15}$ molecules cm$^{-2}$), consistent with the fit errors shown in Fig. 4. Also different absolute

differences are found for the different instruments: MPIC ($\sim -0.53\times10^{15}$ molecules cm$^{-2}$), BIRA ($\sim0.34\times10^{15}$ molecules cm$^{-2}$), and Bremen ($\sim -0.23\times10^{15}$ molecules cm$^{-2}$) display larger differences than the other instruments ($\sim-0.1$ to $\sim0.04\times10^{15}$ molecules cm$^{-2}$). The different absolute differences might be related to possible errors of the elevation angles, interferences of systematic instrumental structures in the fits (e.g., nonlinearity of the detector response and stray light) and differences in the implementations of DOAS fits. For the eight selected days with pronounced diurnal variations of the HONO values,

Figure S5 in the Supplement presents the scatter plots of the HONO delta SCDs with a sequential FRS for each instrument against the reference values. The results of the linear regressions and the linear correlation coefficients are displayed in each subfigure of Fig. S5. As can be seen for 1° elevation angle, all instruments agree well: the scatter plots show compact correlations with correlation coefficients mostly larger than 0.95 (a lower value of 0.86 is only found for the MPIC instrument); the slopes are close to unity with deviations smaller than 16% and intercepts smaller than $0.5\times10^{15}$ molecules

15   cm$^{-2}$. As seen in Fig. 5, smaller correlation coefficients and larger deviations of the slopes and intercepts are found for large elevation angles due to the rather low values and the small range of HONO delta SCDs.

For the HONO delta SCDs derived from fits with a daily noon FRS, we follow the same comparison procedures as for the HONO delta SCDs from fits with a sequential FRS. All five parameters shown in Fig. 5 are quite similar to the results for the sequential FRS for all the instruments. Only the slopes for 15° elevation angle are different, but this phenomenon is due to

the low HONO delta SCDs and small value ranges. To directly show the agreement of the HONO delta SCDs from the fits with the two types of FRS, the mean biases and standard deviations as well as the correlation coefficients, slopes and intercepts of linear regressions derived from the comparisons of two HONO delta SCDs are presented in Fig. 6. And the corresponding histograms of absolute differences between them and their scatter plots are presented in Supplementary Figs. S6 and S7, respectively. For each instrument and each elevation angle, there are no significant mean differences ($<\pm$

$0.04\times10^{15}$ molecules cm$^{-2}$) and standard deviations ($<0.23\times10^{15}$ molecules cm$^{-2}$). The correlation coefficients ($> 0.92$) and slopes (deviations $< 13\%$) are quite close to unity for all the instruments. Only moderate deterioration of correlation coefficients and slopes for the 15° elevation angle are found for some of the instruments.

For the HONO dSCDs derived from fits with a daily noon FRS shown in Fig. 5, the standard deviations are slightly larger than those for the comparisons of the two HONO delta SCDs. This could be caused by the different HONO absorptions in

the daily noon FRS of the different instruments and interferences by the stratospheric species, e.g. ozone. The correlation coefficients are mostly slightly better than for HONO delta SCDs (except for the BIRA instrument) probably due to the slightly larger values of the HONO dSCDs especially for high elevation angles. For the off-zenith observations, Bremen, AIOFM and MPIC have similar mean differences and intercepts for the HONO dSCDs as those for the HONO delta SCDs, while Heidelberg and BIRA show larger and smaller values. This finding is probably caused by differences of the HONO




dSCDs for zenith view between the different instruments. For the CMA instrument, its agreement with the other instruments is better for the HONO dSCDs than for the HONO delta SCDs. The reason could be an unknown problem of the zenith observations of the CMA instrument.

## 3.4 Synthetic spectra and inter-comparisons

In general it is difficult to quantify the biases of the retrieved HONO dSCDs with respect to the real atmospheric values for real MAX-DOAS measurements as the true HONO column is not known. Thus, to assess these biases in more detail, we generated a set of synthetic spectra using the RTM SCIATRAN, version 3.6.0 (03 Dec 2015), in a pseudo-spherical atmosphere (Rozanov et al., 2014) for the same measurement geometries (elevation angles and azimuth angle) and similar sun geometries (16 combinations of SZA and solar azimuth angle (SAA)) as the real measurements. Detailed information on

the RTM simulations is given in Section 1 of the supplement. The simulated delta SCDs at 355nm corresponding to the synthetic spectra are in the range 0.4 to $6\times10^{15}$ molecules cm$^{-2}$ (see Supplementary Fig. S8b), covering the range of values of the real measurements (see Fig. 2a). Note that there is no random noise added into the synthetic spectra.

Two versions of synthetic spectra were generated with different input of water vapour cross sections. The $H_2O$ cross section from the HITRAN 2012 (Rothman et al., 2013) database and from the newly published POKAZATEL line lists (Polyansky

et al., 2016) are used by the RTM to generate version (V) 1 and 2 of the synthetic spectra, respectively. Note that absorption structures below 388 nm exist in the POKAZATEL $H_2O$ cross section, but not in HITRAN. Thus actually there is no $H_2O$ absorption included in the UV range (used in this study) of V1 synthetic spectra. The POKAZATEL $H_2O$ absorption around 363 nm was recently identified in MAX-DOAS and Long-path (LP)-DOAS measurements and could impact the HONO retrieval (Lampel el al., 2016). And they also found that the POKAZATEL line lists underestimate the real $H_2O$ cross section

by a factor of about 2.6. Thus the POKAZATEL $H_2O$ cross section multiplied by 2.6 is used in the RTM. Both the V1 and V2 synthetic spectra are used in the sensitivity studies presented in Section 4, while only the V1 data set is used for the inter-comparison activities.

Six institutes analysed the V1 synthetic spectra using their respective fit software (see Tab. 1). The DOAS settings are almost the same as those for the inter-comparison of the real measurements presented in Tab. 2. The only difference is that

the retrievals are implemented with and without including an intensity offset in the fits. Analyses are performed using a noon (for SAA of 166°) and a sequential FRS. The four settings of the DOAS fits are listed in Tab. 3. Note that only a constant term is used for the setting of including intensity offset correction in the fit, because of the negligible impact on the analyses when including a linear term. Figure 7 shows an example of DOAS fit of the V1 synthetic spectrum for a SAA of 166 ° and EA of 1 ° using the setting #4 in Tab. 3. The retrieved optical depths of the relevant species are comparable to those for the

real measurement shown in Fig. 2a. And the residual structure is smaller than the half of that for the real measurement due to the absence of random noise in the synthetic spectrum. We did the comparisons between the results from the different institutes for the HONO delta SCDs (all the four fit settings) and for the HONO dSCDs (only setting #1 and #2 (noon FRS)). The mean biases and standard deviations as well as the correlation coefficients, slopes and intercepts of the linear regressions





derived from the comparisons of HONO dSCDs (noon FRS) and delta SCDs of different groups with respect to the simulated real values as are presented in Fig. 8. The comparison results are plotted against labels of the different DOAS settings in Tab. 3. And the corresponding histograms of the absolute differences and scatter plots are also shown in Supplementary Figs. S9 and S10, respectively. In general, much larger mean absolute differences for the dSCDs than for the

delta SCDs are found, meanwhile much lower correlations are found for the HONO dSCDs than for the delta SCDs, mainly due to the interference of stratospheric species, e.g. ozone. Correlation coefficients (> 0.91) for the HONO delta SCDs are close to unity for all the groups. The similar mean absolute differences and slopes of HONO delta SCDs between setting #1 and #4 as well as between setting #2 and #3 indicate that the effect of using different FRS on the HONO delta SCDs is negligible for all the groups. However, the effect of intensity offset correction (comparisons between setting #1 and #2 as

well as between setting #3 and #4) on the HONO delta SCDs is found to be considerable (about 0.3 to $0.7 \times 10^{15}$ molecules $cm^{-2}$) for all the groups. The smallest mean absolute differences of the HONO delta SCDs with respect to the real values are smaller than $0.23 \times 10^{15}$ molecules $cm^{-2}$, which are found for setting #1 and #4 (without intensity offset correction) for BIRA, Bremen, AIOFM, MPIC and CMA, and for setting #2 and #3 (with intensity offset correction) for Heidelberg and INTA. The different phenomenon of the intensity offset effect on HONO delta SCD between the two groups of institutes might be

caused by differences in the implementation of intensity offset correction in the DOAS fit software codes. Peters et al. (2016) already demonstrated that different linear fit approach of the intensity offset correction implemented in the DOAS fit can considerably impact the retrieved dSCDs. But nonlinear fit used in QDOAS, WINDOAS and MDOAS was not included in their study. The difference of the nonlinear and linear fit of the intensity offset correction could also be considerable. However, apart from the effect of intensity offset correction by excluding Heidelberg and INTA, the systematic difference of

HONO delta SCDs between the groups with the same DOAS setting is smaller than $0.3 \times 10^{15}$ molecules $cm^{-2}$.

## 4 Sensitivity studies

In this Section we perform sensitivity studies to assess the systematic effect of the absorptions of $H_2O$, $O_4$ and $NO_2$, the Ring spectrum, polynomial, intensity offset and shift corrections on the HONO delta SCD retrievals. We also evaluate the effect of variations of the instrument properties including the wavelength calibration, the instrumental resolution and random noise.

The studies are implemented on both the V1 and V2 synthetic spectra. In addition measurements of the AIOFM instrument on two days of 16 and 18 June, 2013 are analysed, which were selected because of the low and high HONO delta SCDs observed on the two days, respectively. The WINDOAS software is used to implement DOAS fits in the study. And a sequential FRS is used in the DOAS fits.

### 4.1 Residual around 363 nm and the effect of the $H_2O$ absorption in the UV spectral range

In the baseline fit of HONO, a systematic large residual structure around 363nm was found as shown in Fig. 9. If the fit spectral range extends to 390 nm, the residual structure becomes more prominent. Lampel el al. (2016a) demonstrated that a



considerable H$_2$O absorption can be found in MAX-DOAS observations around 363nm. They also showed that the POKAZATEL H$_2$O cross section (Polyansky et al., 2016) can well represent this absorption structure. In Fig. 10a for a measured spectrum by the AIOFM instrument, the residual structures from the fits with and without the POKAZATEL H$_2$O cross section are compared. Especially for the large fit range the residual structures around 363 nm can be minimised by

including the POKAZATEL H$_2$O cross section. The corresponding fit results of the H$_2$O absorptions are also shown in Fig. 10a. In Fig. 10b the corresponding results for the fits of the V2 synthetic spectra are shown. Compared to the results of the measured spectra, the residuals do not contain noise, and the improvement of the residual for the fits by including the H$_2$O cross section becomes even more obvious. The fit results of the V1 synthetic spectra, in which no H$_2$O absorption below 388nm is included, are also shown in Fig. 10c for the comparison with those of the V2 synthetic spectra in Fig. 10b. The

effect of including H$_2$O cross section on the fit residual and the artificially fitted H$_2$O absorption are quite low. Scatter plots of the differences of the HONO delta SCDs from fits with and without including the H$_2$O cross section against the fitted H$_2$O delta SCDs are shown in Supplementary Fig. S11a for the selected AIOFM measurements on 16 and 18 June, 2013. The results are shown for the two fit ranges 335-373 nm and 335-390 nm, respectively. Almost perfect linear correlations are found. An increase of the HONO delta SCDs by up to $1.5 \times 10^{15}$ molecules cm$^{-2}$ is found with respect to the H$_2$O delta SCD

of about $6 \times 10^{23}$ molecules cm$^{-2}$ (in 335-373 nm range) if the H$_2$O cross section is included in the fits. As for the real measurements, similar scatter plots of the changes of the HONO delta SCDs against the fitted H$_2$O delta SCDs for the V2 and V1 synthetic spectra are shown in Supplementary Fig. S11b and c, respectively. Linear correlation coefficients are close to unity and the slopes are similar with those for the measured spectra shown in Fig. S11a.These findings demonstrate that the H$_2$O absorption could mainly contribute to the residual structure around 363nm and can considerably interfere with the

fits of HONO absorption. Thus we conclude that the POKAZATEL H$_2$O cross section should be included in the DOAS fits. However it also needs to be noted that the effect of including H$_2$O cross section on the HONO delta SCDs is found not only for the V2 synthetic spectra ( with UV H$_2$O absorption), but also for the V1 synthetic spectra(without UV H$_2$O absorption). It indicates a possible spectral interference of the POKAZATEL H$_2$O cross section with the structures of other absorptions, e.g. O$_4$ (also reported in Lampel et al., 2016a). Figure 9 indicates that the absorption peek of H$_2$O around 363nm overlaps with

the O$_4$ structures. Further investigations, improved O$_4$ cross sections and H$_2$O cross sections for UV wavelengths are needed to clarify this hypothesis.  In addition it is important to note that the POKAZATEL H$_2$O cross section scaled by 2.6 is used in the fits for the real measurements and synthetic spectra because of the known underestimation (Lampel el al., 2016a).

## 4.2 Candidate fit spectral ranges and interference species

There are four prominent absorption bands of HONO in the spectral range of 335 nm to 390 nm (see Fig. 11a). Thus fits of

HONO absorptions could be implemented in different spectral ranges covering e.g. two, three or four HONO absorption bands. Note that it is unreasonable to extend to the wavelength range below 335nm as strong ozone absorptions and low signal to noise ratios can significantly deteriorate the retrievals and the magnitude of the differential absorption cross-section of HONO decreases here significantly. We compared the HONO delta SCDs between retrieved in the three spectral ranges



of 335-361 nm, 335-373 nm and 335-390 nm for the V1 synthetic spectra and the selected AIOFM measurements as shown in Supplementary Fig. S12a and b, respectively. The results indicate that the HONO delta SCDs retrieved in the wavelength range 335-361 nm and in 334-390 nm are significantly smaller and larger than those in 335-373 nm, respectively. The differences can not be explained by the wavelength dependence of AMF, since this effect can only cause differences of the

HONO SCDs of up to $0.03 \times 10^{15}$ molecules cm$^{-2}$ on average (see Fig. S13 in the Supplement). Therefore the dependence of the retrieved HONO delta SCDs on the fit ranges can be mainly attributed to spectral interferences of the HONO absorption with other absorption structures or instrumental issues. Typical optical depths of the species (based on the measurements during the whole campaign) included in the HONO retrievals are shown in Fig. 11a. In order to assess the possibility of spectral interferences with HONO, we calculated the correlation coefficients of the cross sections of different species with

HONO. The determined correlation coefficients are then scaled with typical atmospheric optical depths of the respective species to roughly estimate their potential for spectral interferences with the HONO absorption. The results shown in Fig. 11b indicate that the strongest interferences are expected from $NO_2$, $O_4$ and the Ring effect. Their individual effects on the HONO retrieval are discussed in the following Sections 4.3 to 4.5.

## 4.3 Influence of the $O_4$ absorption on the HONO analysis

Lampel el al., 2016a reported considerable differences between the three currently available literature $O_4$ cross sections (Greenblatt et al., 1990, Hermans et al., 1999, and Thalman and Volkamer, 2013). For a typical $O_4$ dSCD of $2 \times 10^{43}$ molecules$^2$ cm$^{-5}$, the optical depths of the differences amount to up to $1 \times 10^{-3}$, which is comparable with typical HONO optical depths of up to $3 \times 10^{-3}$. Considering the known wavelength calibration problem of the Greenblatt $O_4$ cross section (Piters et al., 2012), the other two cross sections are probably the best candidates for DOAS fits. We investigate the effects of

changing the $O_4$ cross sections, in the fits on the HONO delta SCDs for the synthetic spectra and the selected AIOFM measurements on 16 June in Supplementary Fig. S14. Similar diurnal variation of the differences of the HONO delta SCDs between the analyses with the Thalman and the Hermans $O_4$ cross section are found for both the synthetic spectra and the measured spectra (see Fig. S14a in the supplement). Since the synthetic spectra are simulated using the Thalman et al. $O_4$ cross section, this finding indicates that the atmospheric $O_4$ absorption is best described by the Thalman et al. $O_4$ cross

section. In Supplementary Fig. S14b, the differences of the HONO delta SCDs are plotted versus the differences of the $O_4$ delta SCDs. Almost linear relationships are found for both synthetic and measured spectra indicating the direct spectral interference between (errors of the) $O_4$ absorption and the retrieved HONO delta SCDs.

The temperature dependence of the $O_4$ cross section is reported in Thalman and Volkamer (2013). The difference of the $O_4$ cross sections between at 203 K and at 293 K is about 20% around 360nm. The Thalman $O_4$ cross section at 203k

orthogonalised to that at 293k is calculated by the orthogonalisation based on Gram-Schmitt's algorithm and is shown in Fig.9. The prominent structure at 203 K indicates that the temperature dependence of $O_4$ cross section probably interferes with the HONO absorption. Moreover, the overlap of the structures of the temperature dependence of $O_4$ cross section with





the $H_2O$ absorption band around 363 nm indicates the potential interplay of the $O_4$ temperature dependence, the $H_2O$ absorption, and the HONO absorption.

In the DOAS fit it is assumed that the AMF (or atmospheric light path) in a spectral range of the fit is constant. However, it is well known that the light path actually depends on the wavelength (Richter, 1997; Marquard et al., 2000; Puķīte et al., 2010; and references therein). This problem could also play a role for the fit of the $O_4$ absorption in the HONO retrievals. The so-called Taylor series approach (TSA) developed by Puķīte et al. (2010) could approximately solve this problem by including a linear term ($\lambda\sigma_{O_4}$) and a square term ($\sigma_{O_4}^2$) of the $O_4$ cross section in the fit ($\lambda$ and $\sigma_{O_4}$ are the wavelength and absorption cross section of $O_4$, respectively). The two TSA terms of $O_4$ orthogonalised to the $O_4$ cross section are shown in Fig. 9. The interplay of Taylor terms of the $O_4$, the structure of the $O_4$ temperature dependence, and the $H_2O$ absorption could impact the retrieved HONO delta SCDs. To test these interference effects in more detail, we compare the HONO delta SCDs from the fits with six different settings for the $O_4$ absorptions (listed in Tab. 4) for the V1 / V2 synthetic spectra and for the selected AIOFM spectra. In these sensitivity studies all other fit settings are kept unchanged (baseline DOAS settings, but without an intensity offset included for the synthetic spectra). For the synthetic spectra, we calculate the differences of the retrieved HONO delta SCDs using the six $O_4$ settings and three spectral ranges with respect to the real HONO delta SCDs (as used in the calculation of the synthetic spectra) (see Supplementary Fig. S15a). Similar differences, but calculated with respect to the results of the baseline retrieval ($O_4$ setting #1 in 335-373 nm, see Tab. 4), are shown in Supplementary Fig. S15b. Note that these differences can be calculated also for the measured spectra. In general the smallest differences of fitted HONO delta SCDs from the real values are found for the wavelength range 335-373 nm. For this wavelength range, also the variation of the fitted HONO delta SCDs by changing the $O_4$ setting is smallest. Also similar differences are found with respect to the real HONO delta SCDs of the synthetic spectra and the retrieved HONO delta SCDs using the baseline settings. Thus we conclude that the wavelength range 335-373 nm is the best suited spectral range to minimise the $O_4$-related interference effects on the HONO retrievals. Another important finding is that for the wavelength ranges $335 - 373$ nm and $335 - 390$ nm the results for the real measurements and the synthetic spectra are similar. Thus we recommend using one Thalman $O_4$ cross section at 293K in the fits. The variation of the HONO delta SCDs by changing the $O_4$ setting indicates the remaining systematic uncertainty related to the $O_4$ effects.

### 4.4 Influence of the Ring spectrum

The temperature dependence of Ring spectrum can contribute to a difference of optical depth of about $5\times10^{-5}$ $K^{-1}$ around 355nm (with respect to a typical Ring optical depth shown in Fig. 11a) based on the study of Lampel el al., 2016a. For the analysis of absorbers with small optical depths, Lampel el al., 2016a recommends including two Ring spectra representing two different temperatures in the fits. To test the effect of the temperature dependence of the Ring effect on the HONO retrievals, we compare the HONO delta SCDs derived using three different Ring settings (see Tab. 4), which are either a Ring spectrum for 250 K, for 273 K or both of them (one is orthogonalised to the other). The corresponding differences of



the derived HONO delta SCDs with respect to the real delta SCDs (for the synthetic spectra) or with respect to the baseline fit (Ring setting #1 in 335-373 nm; for both synthetic and measured spectra) are shown in Supplementary Fig. S16.

The Ring effect on the retrieved HONO delta SCDs is quite different between for the measured and synthetic spectra in the three spectral ranges, especially for Ring setting #3. Thus we recommend using a Ring spectrum at one temperature in HONO retrievals. Furthermore due to the small difference of HONO delta SCDs between Ring Setting #1 and #2, it is reasonable to arbitrarily select 250 K for the generation of Ring spectrum. The variations of the HONO delta SCDs for different Ring settings indicate the remaining systematic uncertainty related to the Ring effect.

## 4.5 AMF wavelength dependence caused by the $NO_2$ absorption

The optical depth of the $NO_2$ absorption can be large, up to about 0.15, which is much larger than the typical optical depth of HONO (up to 0.003). Similar to $O_4$, wavelength dependence of absorption caused by $NO_2$ is also expected. The TSA method (Puķīte et al., 2010) could also be applied for $NO_2$. We compare the HONO delta SCDs from the three fits with different $NO_2$ settings listed in Tab. 4, which are a) the original $NO_2$ cross section, b) including also the linear Taylor term and c) including also the linear and square Taylor terms. The comparison results for the synthetic and measured spectra are shown in Supplementaroy Fig. S17. The results indicate that the $NO_2$ effect on HONO delta SCDs is negligible in the wavelength range 335-373 nm, but considerable in the other two ranges. Also very consistent results for the synthetic spectra and measured spectra are found. Reduction of residual spectral structures related to the $NO_2$ absorption by a use of the TSA method in DOAS fits can be found in the three wavelength ranges. Thus to minimise the $NO_2$ effects, we recommend including the two additional Taylor terms of $NO_2$ in the HONO fit.

## 4.6 Influence of the degree of the polynomial

To account for the broad spectral structures, e.g., related to atmospheric scattering processes, polynomial fits are included in DOAS retrievals. The polynomial degree is usually chosen depending on the spectral range and spectral characteristics of the target species. To quantify the uncertainty of the retrieved HONO delta SCDs related to the choice of the degree of the polynomial, we compare the HONO delta SCDs retrieved by the three fits with different degree of the polynomial (see Tab. 4), including degree 3, 4 and 5. The results for the synthetic and measured spectra in the three spectral ranges are shown in Fig. S18 in the Supplement. The variation of the HONO delta SCDs for different polynomial degrees is smaller in the wavelength range 335-373 nm than in the other two spectral ranges. Also the deviation of the retrieved HONO delta SCDs from the real delta SCDs is generally smallest in the wavelength range 335-373nm. Thus the wavelength range 335-373nm is the best suited spectral range to minimise the polynomial-related uncertainty of HONO retrievals. And the small polynomial effects in the wavelength range 335-373 nm allows to arbitrarily select a fifth degree polynomial for HONO retrievals.



### 4.7 Effect of the intensity offset

To compensate for additional artificial intensity signals like instrumental stray light or insufficient corrections of the dark current or electric offset, an intensity offset correction is normally included in the DOAS fits. Considerable interferences of the intensity offset with the retrievals of trace gases, especially with small optical depths, were reported e.g. by Coburn et al.

(2011). To test the effect of the intensity offset on the HONO analysis, we compare the HONO delta SCDs for different degrees of polynomial for the intensity offset correction (see Tab. 4), including fits without an offset correction and with polynomials of degree 0, 1 and 2 for offset correction. The intensity offset correction is implemented by a non-liner fit in WINDOAS software (Fayt and van Roozendael, 2009). The results for the synthetic and measured spectra in the three spectral ranges are shown in Supplementary Fig. S19. Significant changes of the HONO delta SCDs by including an

intensity offset compared to a fit without an intensity offset are found for both synthetic and measured spectra. Because the intensity offset is expected to be zero for the synthetic spectra, the retrieved non-zero intensity offsets and their influence on the HONO delta SCDs imply a significant interference with HONO retrievals. In spite of these possible interferences, taking into account typical instrumental problems (like spectrograph straylight), the consideration of an intensity offset correction in the fit is still recommended for the HONO retrieval. However, it should be noted that the interference between the fitted

intensity offset and the retrieved HONO delta SCDs as found for the synthetic spectra constitutes a relevant systematic uncertainty of the HONO retrieval.

### 4.8 Effect of the instrumental function and calibration

Supplementary Fig. S3c shows changes of the wavelength calibration of up to 0.015 nm during one day. In this Section the effect of changes of wavelength calibration on the HONO delta SCDs is tested. The tests are done for either excluding or

20 including a wavelength shift in the fit. In a particular test, we manually shifted the synthetic spectra by 0.025 nm. The HONO delta SCDs derived from the shifted spectra are compared to those derived from the original spectra. The differences are only considerable for the fits not including the wavelength shift correction in the fit (leading to differences of 0.2 to 0.4 $\times 10^{15}$ molecules cm$^{-2}$, in the three spectral ranges). The differences are negligible once the shift correction is accounted for in the fit.

In addition changes of the instrumental function could occur. We test the effect of changes of the instrumental function on the HONO retrieval using the synthetic spectra. The cross sections were convoluted with a wrong Gaussian instrumental function with a FWHM of 0.525 nm (instead of 0.50 nm). Then we analysed the synthetic spectra with the new convoluted cross sections. The HONO delta SCDs derived from the new fits are compared with those using the correct instrumental function. The systematic differences are only around -0.02 to -0.13 $\times 10^{15}$ molecules cm$^{-2}$. Here it should be noted that actual

changes of the instrument function are usually smaller than assumed in this test. For example, a change of only 0.004 nm is found for the AIOFM instrument during the whole comparison period. Thus we conclude that the changes of the instrument





function are usually not important for the HONO analysis. But it needs to be noted that asymmetric changes and wavelength dependence changes of the instrumental function are not considered in the test study.

### 4.9 Effect of random noise

The measured spectra are subject to several sources of random noises (i.e., photon noise or electronic noise). To quantify the effect of noise on the HONO analysis, Gaussian random noise with a signal to noise ratio (SNR) of 3000 is added into the V1 synthetic spectra. We compare the HONO delta SCDs and the fit errors of the synthetic spectra with noise and without noise. The comparison results are shown in Fig. 12. The results indicate that the fit errors increase from around $0.1 \times 10^{15}$ molecules cm$^{-2}$ for spectra without noise to ~$0.24 \times 10^{15}$ molecules cm$^{-2}$ for the noisy spectra. The largest increase of the fit error is found for the wavelength range 335-361 nm. However, it should be noted that in the spectral range of 335-373nm the fit error for the synthetic spectra with noise is rather low (about $0.15 \times 10^{15}$ molecules cm$^{-2}$), which is similar to that of the real measurements of the best instruments as shown in Fig. 4. We find no considerable systematic effect of noise on the HONO delta SCDs. However, the standard deviations of the HONO delta SCDs for the spectra either including or excluding noise are considerable and in the range of $0.12 \times 10^{15}$ molecules cm$^{-2}$ to $0.22 \times 10^{15}$ molecules cm$^{-2}$. The largest standard deviation is found in the wavelength range 335-361 nm.

### 5. Recommended analysis settings and error budget

Systematic uncertainties of the HONO retrieval related to the different error sources in the three spectral ranges are summarized in Fig. 13a based on the sensitivity studies presented in Section 4. In addition to these errors, the error of the HONO cross section is estimated as 5% (Stutz et al., 2000). Fig. 13a indicates that the uncertainties related to the intensity offset fit, the $O_4$ and $H_2O$ absorptions, and the Ring effect are usually the prominent errors sources. Another important finding is that the spectral range of 335-373 nm has the lowest systematic uncertainty. Systematic biases of the retrieved HONO delta SCDs for the synthetic spectra compared to the real values and random uncertainties (corresponding to the noise of SNR of 3000) are shown in Fig. 13b for the three spectral ranges. Smallest systematic and random uncertainties are again found for the spectral range of 335-373 nm with a random uncertainty typically smaller than 25% of the systematic uncertainty. Therefore we recommend to retrieve HONO in the spectral range of 335-373nm. In addition, as discussed in Section 4.1, the POKAZATEL $H_2O$ cross section is suggested to be included in HONO retrievals in 335-373nm. The other fit settings should be kept as they are in the baseline DOAS setting (see Tab. 2).

### 6 Conclusions

HONO dSCDs and delta SCDs derived from the seven MAX-DOAS instruments during the MAD-CAT campaign held in Mainz were systematically compared. The fit errors of the HONO dSCDs derived from the instruments with cooled large-





size detectors were found to be in the range of about 0.1 to $0.3 \times 10^{15}$ molecules cm$^{-2}$ for an integration time of one minute, while the fit error for the mini MAX-DOAS instrument is around $0.7 \times 10^{15}$ molecules cm$^{-2}$. Although the HONO delta SCDs (the difference of the HONO SCDs for the non-zenith observations and the zenith observation of the same elevation sequence) are usually smaller than $6 \times 10^{15}$ molecules cm$^{-2}$, time series of the HONO delta SCDs retrieved from different instruments are well consistent. Similar consistent results are found for the fits with a sequential FRS and a daily noon FRS. Except for the mini-MAX-DOAS instrument, the systematic absolute differences of the HONO delta SCDs between the instruments are smaller than $0.63 \times 10^{15}$ molecules cm$^{-2}$, while the standard deviations are smaller than $0.68 \times 10^{15}$ molecules cm$^{-2}$. The correlation coefficients are higher than 0.7 and the slopes of linear regressions deviate from unity by less than 16% for the elevation angle of 1 °, but the correlations decrease with increasing elevation angles. All instruments can well observe the temporal variation of the HONO delta SCDs for low elevation angles. Furthermore, there are no considerable systematic differences of the HONO delta SCDs from the fits with the sequential FRS and the daily noon FRS for all the instruments except the mini MAX-DOAS instrument. The standard deviations are lower than $0.23 \times 10^{15}$ molecules cm$^{-2}$. In addition the deviations of the HONO dSCDs derived from the fits with daily noon FRS between the instruments are generally larger than those of the HONO delta SCDs mainly due to the different HONO absorptions in the noon FRS and the interferences by the stratospheric species, e.g. ozone.

We evaluated the consistency of the DOAS fits by the different groups by using synthetic spectra, for which the real HONO dSCD and delta SCDs are known. The differences of the HONO dSCDs from the real values are much larger than those of the HONO delta SCDs for all groups mainly due to the interferences by the stratospheric species. The smallest differences ($<0.23 \times 10^{15}$ molecules cm$^{-2}$) of the HONO delta SCDs from the real values are found for the DOAS settings without the intensity offset correction for most groups, but for two groups the smallest differences are found if the intensity offset correction was included. The different effect of the intensity offset correction might be due to the different implementation of intensity offset correction in the software codes of DOAS fits. However, apart from the effect of intensity offset correction, the systematic differences of HONO delta SCDs for the synthetic spectra between the groups (caused by implementation of DOAS fits in the software packages) are smaller than $0.3 \times 10^{15}$ molecules cm$^{-2}$, about half of the systematic differences of the real measurements between the different instruments.

We compared the HONO delta SCDs obtained from fits with a sequential FRS in three spectral ranges (335-361 nm, 335-373 nm and 335-390 nm) and found significant differences. The HONO delta SCDs in the wavelength ranges 335-361 nm and 335-390 nm are systematically different from those in the wavelength range 335-373nm by $-0.08 \times 10^{15}$ molecules cm$^{-2}$ and $+0.57 \times 10^{15}$ molecules cm$^{-2}$, respectively. To characterize the dominant systematic error sources and to find the best suited DOAS settings for the HONO analysis, we performed various sensitivity studies based on the synthetic spectra and selected measurements from the AIOFM instrument. The main findings are listed below:

1)  Systematic residual structures are found around 363 nm, which are most probably caused by the H$_2$O absorption around this wavelength. Meanwhile if the POKAZATEL H$_2$O cross section is included in the spectral analysis, a systematic





increase of the HONO delta SCDs of up to $1.5 \times 10^{15}$ molecules $cm^{-2}$ is found. Because of the two phenomenon, we recommend including the POKAZATEL $H_2O$ cross section in the fits. The uncertainty caused by the potential interference of the absorption of $H_2O$ and other species (in particular $O_4$) with the HONO absorption is found to be about $0.13 \times 10^{15}$ molecules $cm^{-2}$ in the wavelength range 335-373 nm and $0.5 \times 10^{15}$ molecules $cm^{-2}$ in the wavelength range 335-390 nm.

2) We investigated further potential interferences with all spectral structures included in the HONO analysis and found that the strong effects also from interferences of $NO_2$, $O_4$, and the Ring spectrum.

3) Analysis results using different $O_4$ cross sections indicated that the $O_4$ Thalman cross section describes the real atmospheric $O_4$ absorptions best and should be used in the HONO analysis. Systematic uncertainties related to the wavelength dependence of the AMF caused by the $O_4$ absorptions and its temperature dependence are about $0.5 \times 10^{15}$ molecules $cm^{-2}$ in the wavelength range 335-361 nm, $0.1 \times 10^{15}$ molecules $cm^{-2}$ in the wavelength range 335-373 nm and $0.2 \times 10^{15}$ molecules $cm^{-2}$ in the wavelength range 335-390 nm.

4) The uncertainties related to the temperature dependence of Ring effect are about $0.35 \times 10^{15}$ molecules $cm^{-2}$ in the wavelength range 335-361 nm, $0.2 \times 10^{15}$ molecules $cm^{-2}$ in the wavelength range 335-373 nm and $0.12 \times 10^{15}$ molecules $cm^{-2}$ in the wavelength range 335-390 nm. However, the results of the sensitivity tests are not completely conclusive, thus we still recommend to simply using a Ring spectrum only for one temperature in the HONO analysis.

5) We also investigated the wavelength dependence of the AMF caused by the $NO_2$ absorption. We found that the effect on the HONO retrievals can be well compensated by the Taylor series approach from Puķīte et al. (2010). Thus we suggest including the linear and square Taylor terms in the HONO analysis.

6) The systematic uncertainties related to the choice of the polynomial are about $0.2 \times 10^{15}$ molecules $cm^{-2}$ in the wavelength range 335-361 nm, $0.04 \times 10^{15}$ molecules $cm^{-2}$ in the wavelength range 335-373 nm and $0.25 \times 10^{15}$ molecules $cm^{-2}$ in the wavelength range 335-390 nm. Thus we conclude that the spectral range 335-373nm, is the best choice to minimize the influence of the choice of the polynomial on the HONO results.

7) The systematic uncertainties related to the intensity offset are about $0.55 \times 10^{15}$ molecules $cm^{-2}$ in the wavelength range 335-361 nm, $0.35 \times 10^{15}$ molecules $cm^{-2}$ in the wavelength range 335-373 nm, and $0.25 \times 10^{15}$ molecules $cm^{-2}$ in the wavelength range 335-390 nm. Although the results from the synthetic spectra (which are not subject to any artificial offsets) indicate a systematic interference between the fitted intensity offset and the retrieved HONO delta SCDs, we still recommend including the intensity offset in the fit, because for real measurements it can correct instrumental shortcomings like spectrograph straylight.

8) Variations of the instrumental wavelength calibration, the instrument slit function, and random noise have only little contribution to the systematic uncertainties of the HONO retrievals.

In summary we find that the total systematic uncertainty from the different error sources is much smaller in the spectral range 335-373 nm ($0.87 \times 10^{15}$ molecules $cm^{-2}$) compared to that in the other two investigated spectral ranges. Moreover, the systematic bias of the measured HONO delta SCDs from the simulated real values in the synthetic spectra are also smallest



in the wavelength range 335-373nm (about $0.02 \times 10^{15}$ molecules cm$^{-2}$). Thus 335-373 nm is the recommended fit range for HONO retrievals.

In this spectral range, the typical random uncertainty is about $0.16 \times 10^{15}$ molecules cm$^{-2}$, which is only 25% of the total systematic uncertainty. These results are obtained for an assumed SNR of 3000, which is close to what the best instruments

5    considered achieved in this study. As a final result we conclude that most of the MAX-DOAS instruments can well observe atmospheric HONO absorptions in situations with HONO delta SCDs higher than $0.2 \times 10^{15}$ molecules cm$^{-2}$. Further work should aim to better quantify the spectral interferences between the absorptions of HONO and other absorbers in the selected spectral range. Also further studies on the interference between the HONO absorption and the intensity offset correction are recommended.

**Acknowledgements:** This work was supported by Max Planck Society-Chinese Academy of Sciences Joint Doctoral Promotion Programme and National Natural Science Foundation of China (Grant No.: 41275038 and 41530644).

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





# Figures

(a) (b)

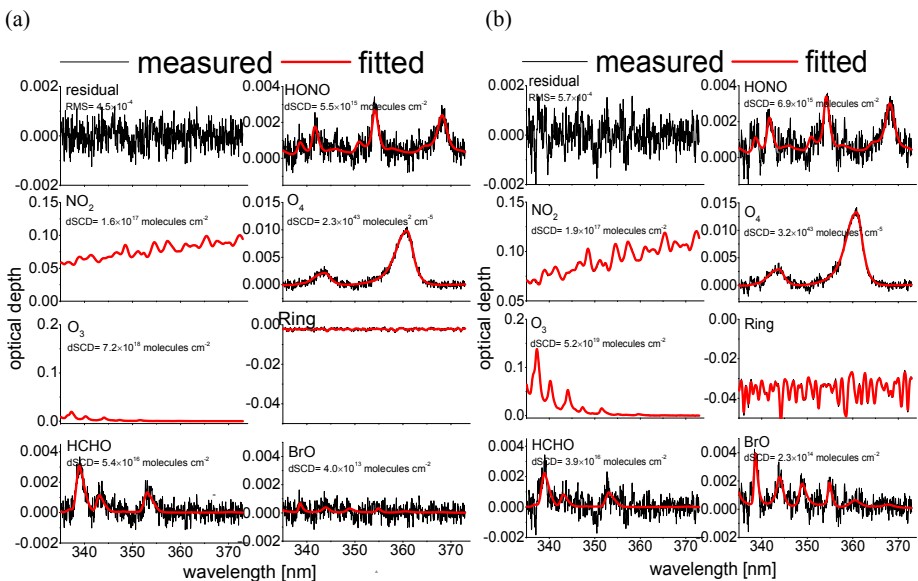

**Figure 1: Examples of HONO fits of a spectrum acquired by the AIOFM instrument at around 04:00 UTC on 18 June 2013 for 1° elevation angle and 50° azimuth angle. A sequential FRS around 03:58 UTC (a) or a noon FRS around 11:30 UTC (b) are respectively used.**

(a)

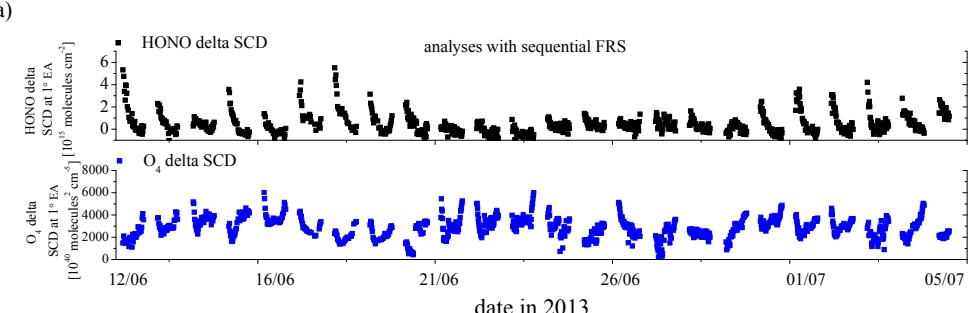

(b)

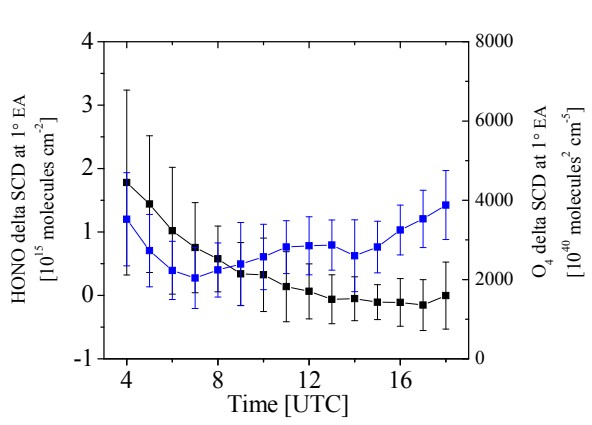



**Figure 2: (a) The hourly averaged HONO and O$_4$ delta SCDs for 1° elevation angle (using a sequential FRS) derived from the measurements of the AIOFM instrument during the whole comparison period. (b) For the same data, the averaged diurnal variations and the respective standard deviations (error bars) in each hour are given.**

### 3 July 2013

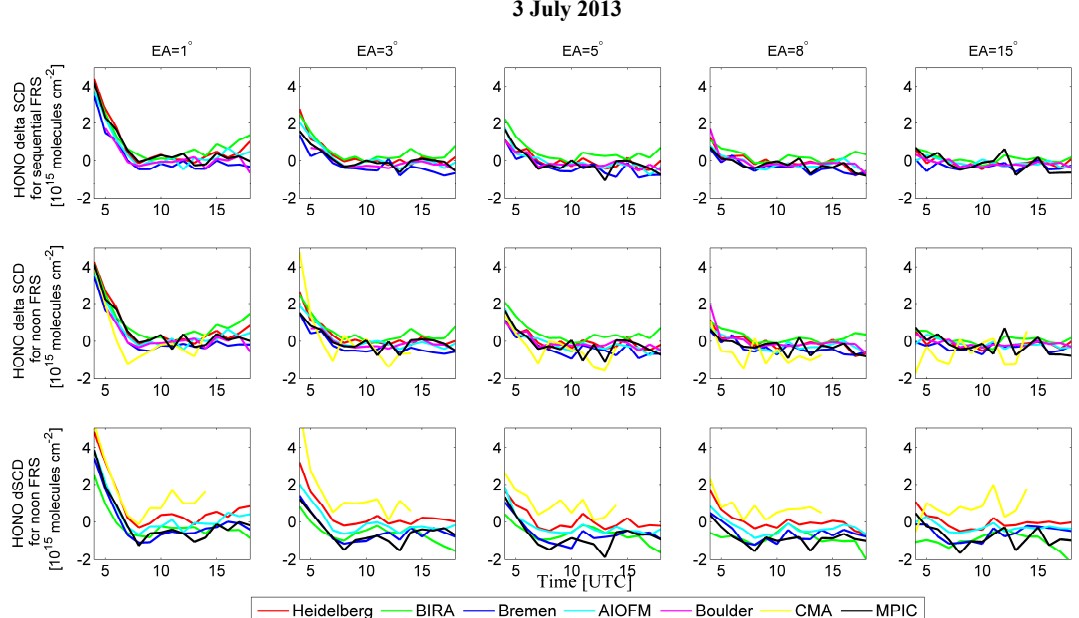

**Figure 3: Time series of the hourly averaged values of HONO delta SCDs using a sequential FRS and a daily noon FRS as well as the HONO dSCDs with a daily noon FRS for different elevation angles and participating instruments on 3 July 2013.**





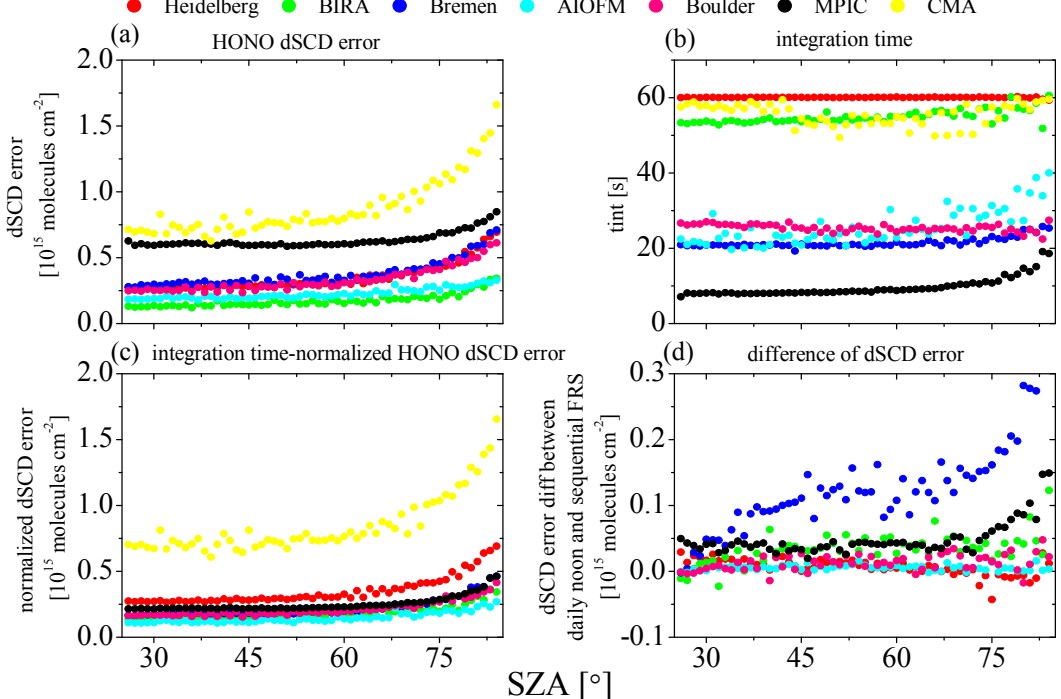

**Figure 4: Comparison of different fit errors as a function of the SZA between participating instruments: a) Averaged HONO dSCD fit errors for spectra at 1 ° elevation angle using a daily noon FRS, b) integration time c) normalized HONO dSCD fit errors according to an integration time of one minute, d) differences of the HONO dSCD fit errors with either a daily noon FRS or a sequential FRS.**



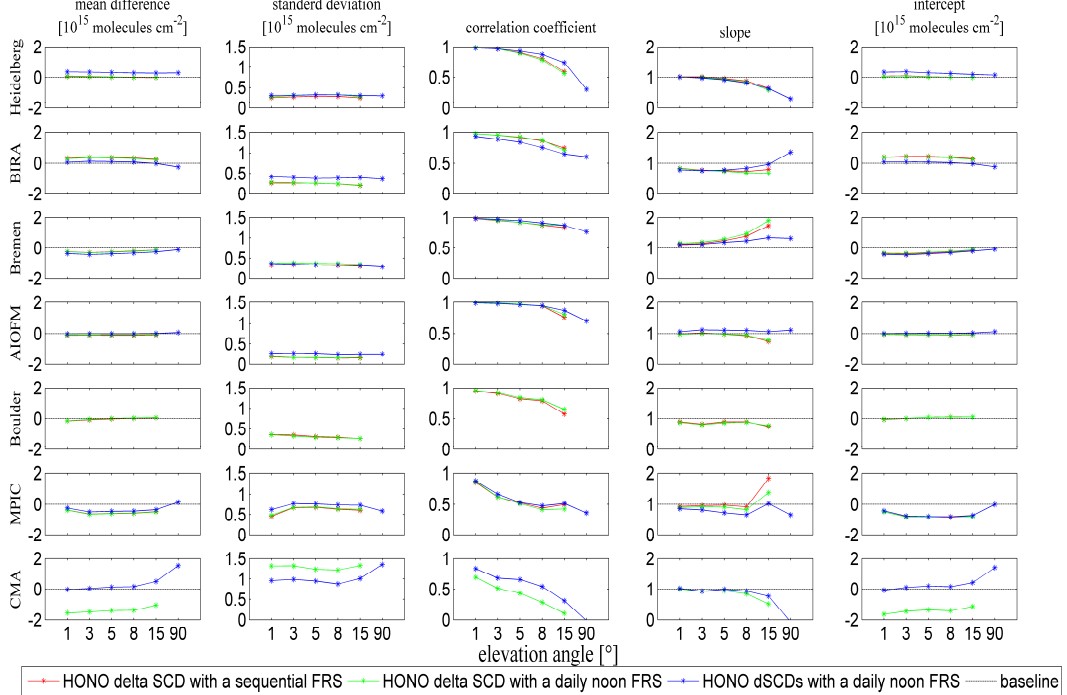

**Figure 5:** Mean differences and standard deviations as well as correlation coefficients, slopes and intercepts of linear regressions derived from comparisons of the HONO delta SCDs and dSCDs retrieved from different instruments with reference values as function of the elevation angle. The HONO delta SCDs are derived from fits with a sequential FRS (red curves) and a daily noon FRS (green curves). The HONO dSCDs are derived from fits with a daily noon FRS(blue curves).

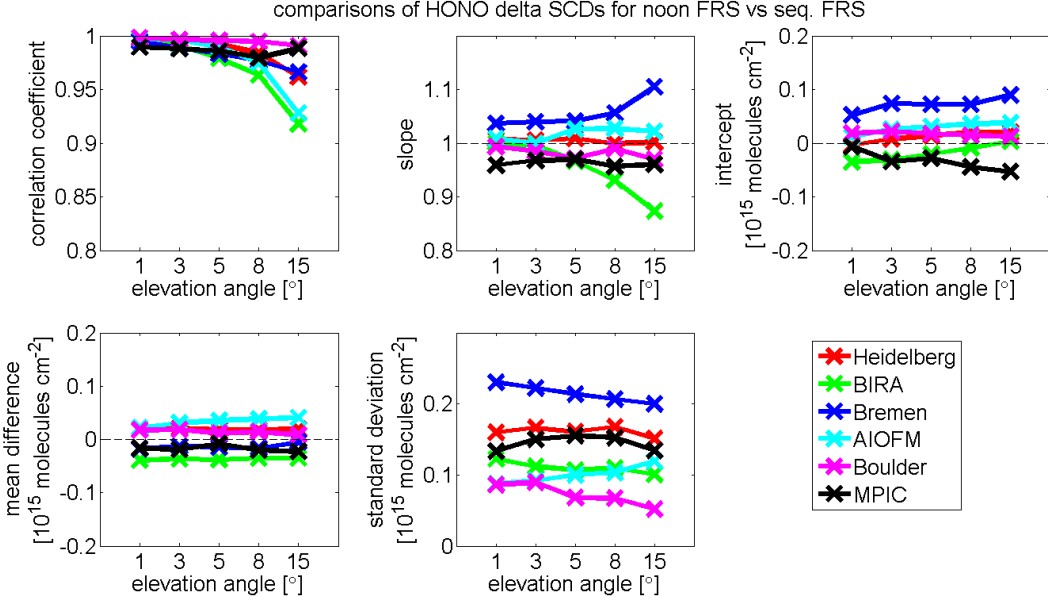

**Figure 6:** correlation coefficients, slopes and intercepts of linear regressions as well as mean differences and standard deviations derived from the comparisons of the HONO delta SCDs retrieved by fits between with a daily noon FRS and a sequential FRS as function of the elevation angle for individual instruments. The color curves indicate different instruments.





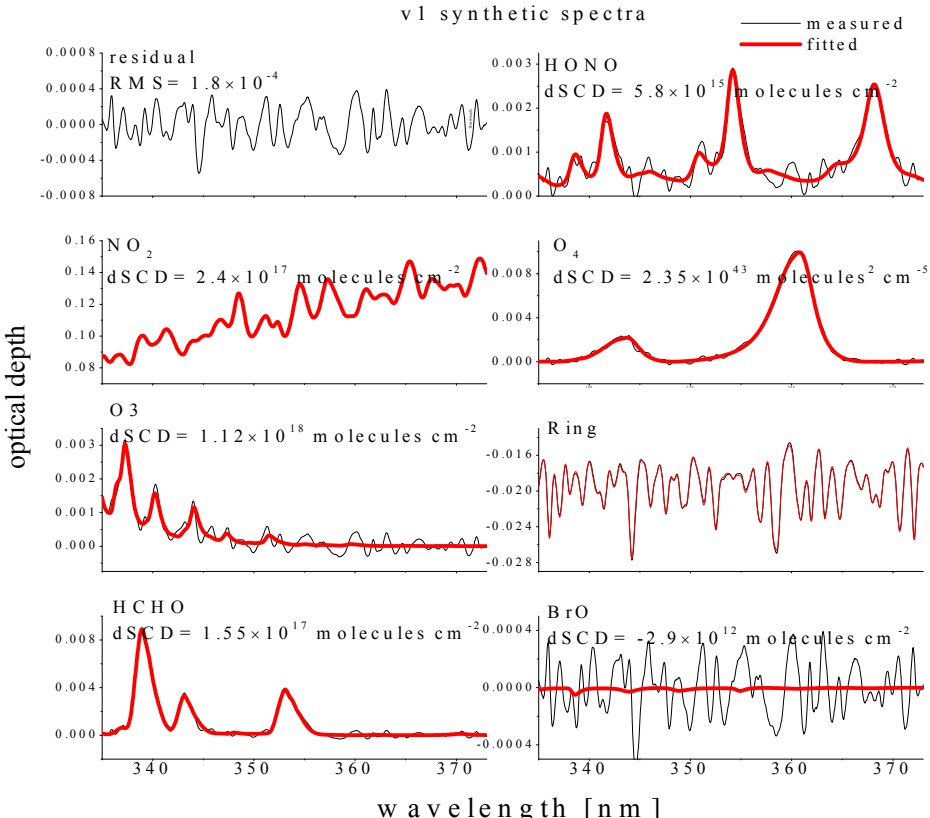

**Figure 7: Example of a HONO fit of a V1 synthetic spectrum for a SAA of 166° and EA of 1° using the DOAS setting with a sequential FRS without the intensity offset correction (setting #4 in Table 4).**



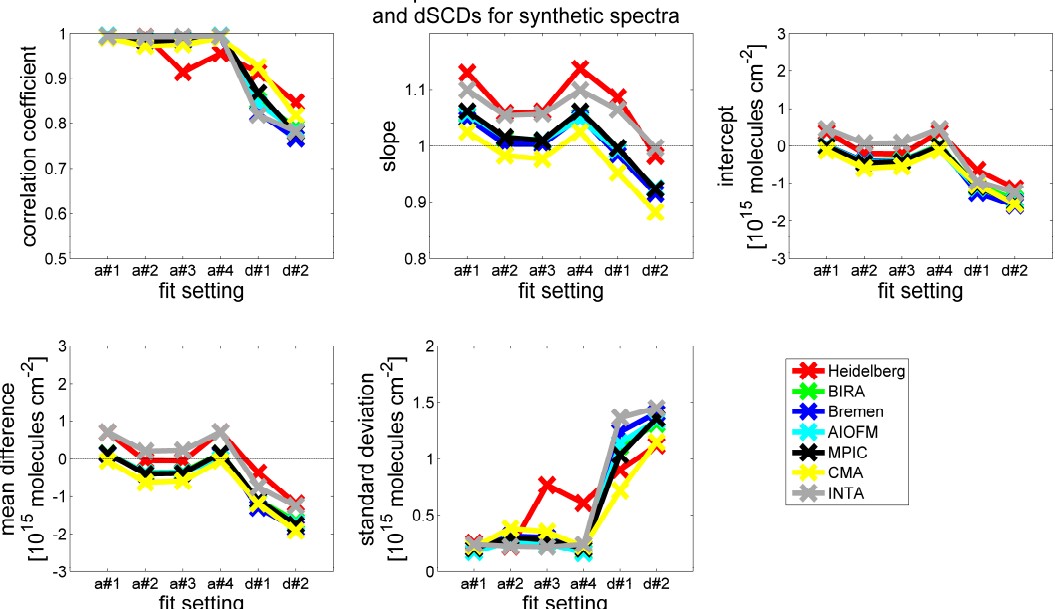

**Figure 8:** correlation coefficients, slopes and intercepts of linear regressions as well as mean differences and standard deviations derived from the comparisons of the HONO delta SCDs and dSCDs between retrieved from the V1 synthetic spectra and the simulated real values as function of the DOAS fit setting number for individual institutes. The color curves indicate different institutes. "a#1", "a#2", "a#3", and "a#4" in the labels of x-axis indicate the HONO delta SCD retrieved by fits with DOAS setting #1, #2, #3 and #4 (see Table 4), respectively. And "d#1" and "d#2" indicate the HONO dSCDs retrieved by fits with DOAS setting #1 and #2, respectively.




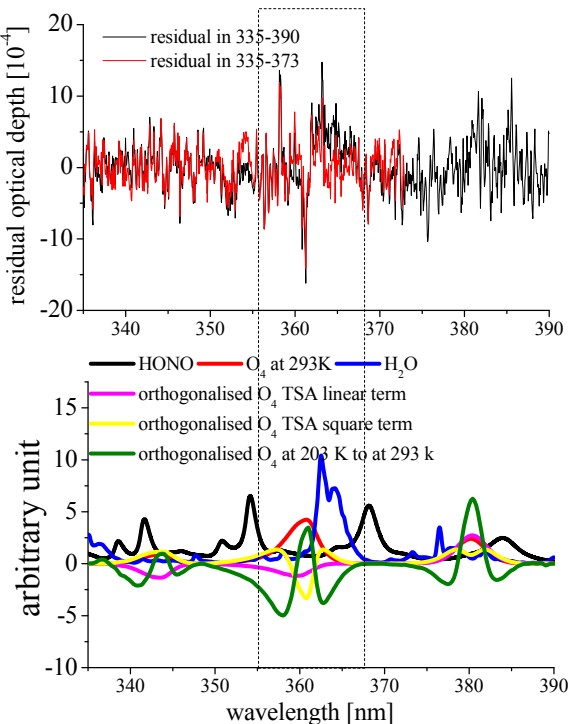

**Figure 9: Top: Residual structure from baseline fits with a sequential FRS of the measured spectrum at 1° elevation angle around noon on 16 June, 2013 in the spectral range of 335-390 nm (black) and 335-373 nm (red). Bottom: normalised absorption cross sections used in the HONO baseline fit.**

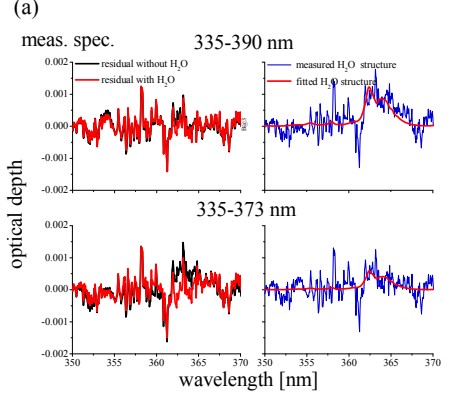

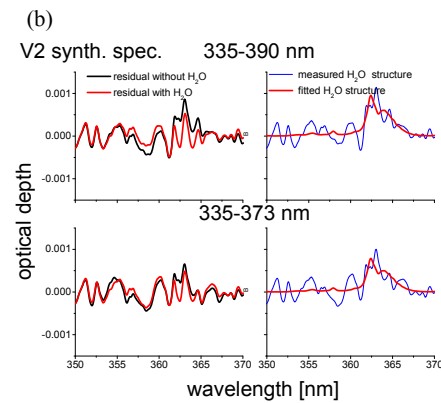




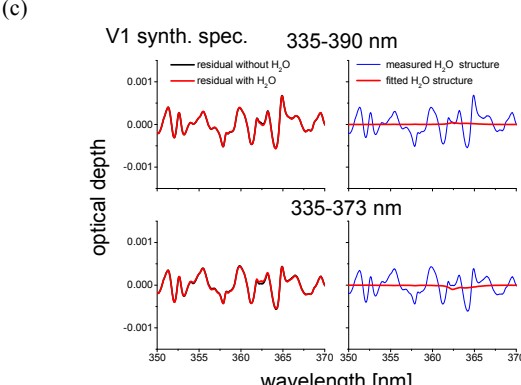

**Figure 10: (a) Residual structures (left) and H₂O fit results (right) for the same measured spectrum as in Fig. 9 for the fits either with (red) or without (black) the POKAZATEL H₂O cross section. The upper and lower two subfigures represent fits in the spectral range of 335-390 nm and 335-373 nm, respectively. (b) and (c) are same as (a), but for the V2 and V1 synthetic spectra for a SAA of 166° and EA of 1°, respectively.**

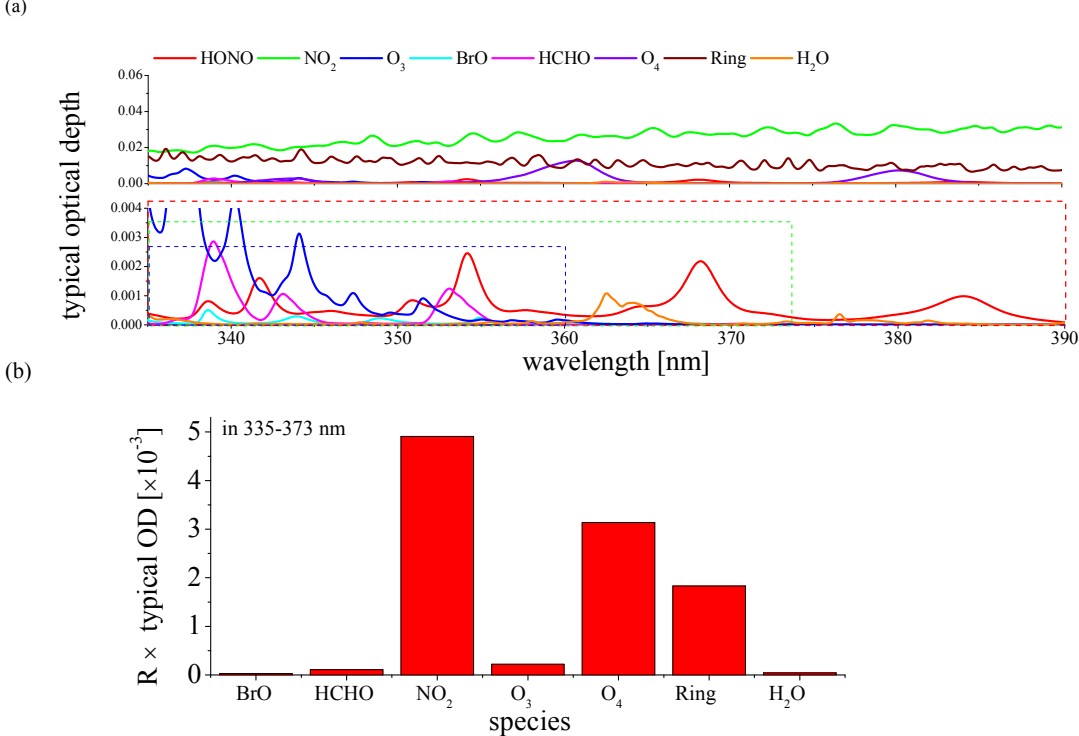

**Figure 11: (a) Typical optical depths of the absorption species as the function of wavelength in the wavelength range for HONO retrieval; the dashed colored squares in the lower subplot indicate the wavelength ranges 335 nm up to 360 nm, 373nm and 390 nm, respectively; (b) Correlation coefficients of the different cross sections multiplied with typical optical depths of respective species (based on the measurements during the whole campaign) with the HONO cross sections in the spectral range 335-373 nm.**




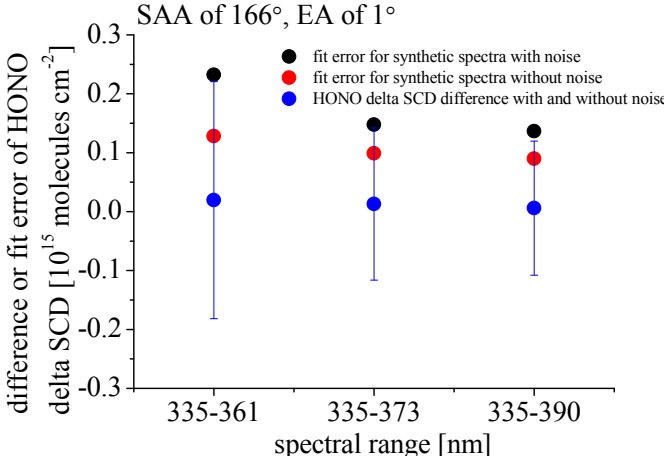

**Figure 12: Comparison of the averaged fit errors of the HONO delta SCDs between for synthetic spectra with and without noise (black and red dots), and the averaged differences of the HONO delta SCDs between derived from the two synthetic spectra (blue dots). The error bars indicate the corresponding standard deviations of the differences.**

(a)                                                        (b)

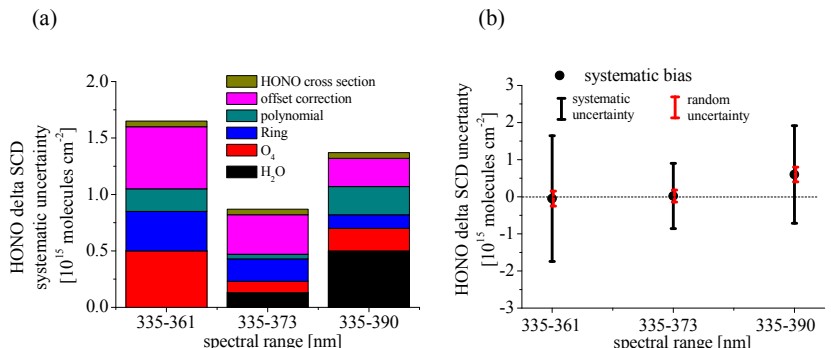

**Figure 13: (a) Systematic uncertainties of the HONO delta SCDs with respect to different error sources for the three spectral ranges. (b) Systematic biases from the real values, of the retrieved HONO delta SCDs (black dots) derived from the synthetic spectra in the three spectral ranges; black and red bars indicate typical total systematic and random uncertainties (for a SNR of 3000) of the retrieved HONO delta SCDs.**





# Tables

Table 1 Overview on instrumental properties and analysis software used by the different institutes participating in the HONO comparison activity

| Institute | Detector Characteristics | Observed wavelengths (nm) | FWHM (nm) | Pixel Sampling (nm) | Integration time per spectrum (s) | Field of view (° FWHM) | Manufacturer | Instrument Reference | Fit software | Inter-comparison activity[e] | |
|---|---|---|---|---|---|---|---|---|---|---|---|
| | | | | | | | | | | Real Meas. | Synth. |
| Heidelberg | AvaSpec-ULS 2048 pixels back-thinned Hamamatsu S11071-1106 CCD | 294-459 | 0.59 at 334 nm | ~0.09 | ~60 | 0.2 | Envimes | Lampel et al. (2015) | DOASIS[1] | × | × |
| BIRA | 2D Back-illuminated CCD, 2048×512 pixels (-40°C) | 300-386 | 0.49 | ~0.04 | ~55 | 0.5 | self-built | Clémer et al., (2010) | QDOAS[2] | × | × |
| Bremen | 2D Back-illuminated CCD, 1340×400 pixels (-35°C) | 308-376 | 0.43 | ~0.05 | ~20 | 0.8 | self-built | Peters et al. (2012) | NLIN[3] | × | × |
| AIOFM[a] | 2D Back-illuminated CCD, 2048×512 pixels (-40°C) | 288-410 | 0.35 | ~0.06 | ~25 | 0.4 | self-built | Wang et al. (2014) | | × | × |
| Boulder | 2D Back-illuminated CCD, 1340x400 pixels (-30°C) | 329-472 | 0.78 | ~0.07 | ~25 | 0.95 | self-built | Ortega et al. (2015, 2016) | WINDOAS[4] | × | |
| MPIC[b] | DV420A-BU, Andor 2D back-illuminated CCD, 1024x255 pixels (-30°) | 319-457 | 0.6-0.8 | ~0.14 | ~10 | 0.6 | self-built | Krautwurst, 2010 | WINDOAS[4] / MDOAS[5] | × (WINDOAS[4]) | × (MDOAS[5]) |
| CMA[c] | 2048 pixel, Sony ILX511 CCD | 292-447 | 0.6-0.8 | | <=60 | 0.8 | Hoffmann Messtechnik GmbH | Jin et al. (2016a and b) | WINDOAS[4] | × | × |
| INTA[d] | - | - | - | - | - | - | - | - | LANA[6] | | × |

a. Anhui Institute of Optics and Fine Mechanics, Chinese Academy of Sciences

b. Max Planck Institute for Chemistry

c. China Meteorological Administration

d. Área de Investigación e Instrumentación Atmosférica, Madrid, Spain.

e. The flag indicates whether the group participates in the inter-comparison activity of the real measurements and the synthetic spectra or not.





1. Reference: Kraus, 2006.

2. Reference: Danckaert et al., 2012.

3: Reference: Richter, 1997.

4. Reference: Fayt and van Roozendael, 2009.

5. Reference: J. Remmers, DOAS fits implemented by MATLAB, personal communication, 2013.

6. Reference: Gil et al., 2008.

Table 2 Baseline DOAS analysis settings of HONO delta SCDs and dSCDs for the comparison activity and recommended settings. √ indicates the item for recommended setting is same as that for baseline setting.

| Parameter | Baseline setting | Recommended setting |
|---|---|---|
| Fitting spectral range | 335-373 nm | √ |
| Wavelength calibration | Calibration based on Fraunhofer lines of Kurucz solar spectrum (Kurucz et al., 1984) | √ |
| Cross sections | | |
| HONO | Stutz et al. (2000), 296 K | √ |
| NO$_2$ | Vandaele et al. (1998), 220 K and 298 K, I$_0$-corrected[*] ($10^{17}$ molecules cm$^{-2}$) Taylor terms (see Pukīte et al. 2010) with respect to $\sigma_{NO_2}$ at 298 K : $\lambda\sigma_{NO_2}$, $\sigma^2_{NO_2}$ | √ |
| O$_3$ | Bogumil et al., (2003), 223 K and 243 K, I$_0$-corrected[*] ($10^{20}$ molecules cm$^{-2}$) | √ |
| BrO | Fleischmann et al. (2004), 223 K | √ |
| O$_4$ | Thalman and Volkamer (2013), 293 K | √ |
| HCHO | Meller and Moortgat (2000), 297 K | √ |
| H$_2$O (vapor) | Not included | Polyansky et al. (2016) scaled by 2.6 (Lampel el al., 2016a) |
| Ring effect | Ring spectrum calculated based on Kurucz solar atlas and Ring scaled with ($\lambda$/ 354 nm)$^4$ (Wagner et al., 2009) | √ |
| Intensity offset | Polynomial of order 1 (corresponding to 2 coefficients) | √ |
| Polynomial term | Polynomial of order 5 (corresponding to 6 coefficients) | √ |
| Wavelength adjustment | All spectra are shifted and stretched against FRS | √ |
| Fraunhofer Reference Spectrum (FRS) | 1. daily noon FRS (at 11:30) 2. sequential FRS | √ |

* solar I$_0$ correction, Aliwell et al., 2002





Table 3 Four DOAS fit settings:

| Setting | Intensity offset fit (constant) | Noon FRS | Sequential FRS |
|---------|--------------------------------|----------|----------------|
| **#1** | No | Yes | No |
| **#2** | Yes | Yes | No |
| **#3** | Yes | No | Yes |
| **#4** | No | No | Yes |

Table 4 DOAS fit settings for the sensitivity studies with respect to $O_4$, Ring, $NO_2$, polynomial and intensity offset correction in Section 4.

| item | type | fit setting |
|------|------|-------------|
| $O_4$ | #1 | $O_4$ at 293 K |
| | #2 | $O_4$ at 293 K + Taylor linear term of $O_4$ |
| | #3 | $O_4$ at 293 K + Taylor linear term of $O_4$ + Taylor square term of $O_4$ |
| | #4 | $O_4$ at 293 K and 203 K |
| | #5 | $O_4$ at 293 K and 203 K + Taylor linear term of $O_4$ at 293 K |
| | #6 | $O_4$ at 293 K and 203 K + Taylor linear term of $O_4$ at 293 K + Taylor square term of $O_4$ at 293 K |
| Ring | #1 | Ring at 250 K |
| | #2 | Ring at 273 K |
| | #3 | Ring at 250 K and 273 K |
| $NO_2$ | #1 | $NO_2$ |
| | #2 | $NO_2$ + Taylor linear term of $NO_2$ |
| | #3 | $NO_2$ + Taylor linear term of $NO_2$ + Taylor square term of $NO_2$ |
| Polynomial | #1 | Polynomial of degree 5 |
| | #2 | Polynomial of degree 4 |
| | #3 | Polynomial of degree 3 |
| offset | #1 | No offset correction |
| | #2 | Polynomial of degree 0 |
| | #3 | Polynomial of degree 1 |
| | #4 | Polynomial of degree 2 |