# Peer review of "MAX-DOAS measurements of HONO slant column densities during the MAD-CAT Campaign: inter-comparison, sensitivity studies on spectral analysis settings, and error budget"

_Atmospheric Measurement Techniques, 2016_

## Referee Comment (RC1) · M. Wenig (Referee) · 3 May 2017

The paper describes a very extensive comparison study of several MAX-DOAS HONO measurements from different groups. Using coinciding measurements in Mainz as well as modeled spectra, their comparison can focus on different aspects of the instrument characteristics and retrieval process. The paper addresses relevant scientific questions since the findings can be used for improving existing retrievals and are very valuable for future instrument and algorithm design. It is more an evaluation of existing methods rather than the introduction of a novel concept, but the methods and conclusions are

very solid. At some points the paper is a little vague, when only speculations are provided instead of explanations for observed differences. The title is appropriate and the abstract summarizes the presented work well. The paper is a little long, especially in combination with the supplementary material. The paper should be readable without the supplement as it should only provide extra material if the reader wants to go into more details. Overall the manuscript is of high scientific quality and I recommend publication after some minor corrections and clarifications as listed in the following:

L13p2 "uncertainty from different sources" sounds confusing, how about "uncertainty from all the different sources" or "overall uncertainty"?

L14p2 "the systematic bias of the fitted from the simulated real HONO delta SCDs is" is grammatically challenging as well.

L18p2 The sentence "However, systematic uncertainties limit the reliability of the results." is very general. What do you know about systematic errors? If the bias is known, you can correct for it. If not, how reliable are the results?

L25ffp5 When you define the different quantities, please provide equations. That would make it easier later on in the paper if you could just refer to the variable. For example what is the difference between delta SCD with sequential FRS and dSCD?

L4ffp6 When you describe Fig. 1 you mention the different fit results for O3, NO2 and BrO but for HONO you just write "The HONO absorption structures are well retrieved using both types of FRS.", but the fit parameters are 5.5 and 6.9 10ˆ15 molec. cmˆ-2, so quite different.

L31fp6 You write "because of unknown instrumental problems, CMA and Boulder didn't participate in the comparisons of the delta SCDs for a sequential FRS and dSCDs for a daily noon FRS, respectively", you could mention that the other measurements were not affected.

L1fp7 In the sentence "all the instruments capture well the diurnal evolution and eleva-

tion angle dependence of the HONO delta SCDs." "capture well" sounds very general, you could be more specific by mentioning if they agree within the uncertainties or if there are significant differences.

L10ffp8 You use the average of selected instruments as the reference values, but then you compare all instrument to that reference (l13p8,l11p9), that's not very consistent. How about using a median or something similar for reference instead?

L2fp10 Again, an "unknown problem" can always be the reason for discrepancies, can you be more spefic?

L12p10 You mention "Note that there is no random noise added into the synthetic spectra." But in Sec. 4.9 (l4fp17,also table S1) a SNR of 3000 is mentioned that is added to the synthetic spectra, so why not in Sec. 3.4?

L29p13 You write "The Thalman O4 cross section at 203k orthogonalised to that at 293k is calculated by the orthogonalisation based on Gram-Schmitt's algorithm and is shown in 30 Fig.9." and Fig. 9's caption says "normalized absorption cross sections". How were they normalized or related to that what scalar product was used for the orthogonalization algorithm, e.g. is a polynomial removal part of the scalar product?

L29p15 The degree of the polynomial was "arbitrarily selected" as five, why not 3? Fewer parameters to fit usually make the optimization more stable.

L12p16 "In spite of these possible interferences, taking into account typical instrumental problems (like spectrograph straylight), the consideration of an intensity offset correction in the fit is still recommended for the HONO retrieval." I agree, but can you underline this statement with some data or estimates? How do you know which influence is more important?

L23fp18 What do you mean exactly with "systematic differences [. . .] caused by implementation of DOAS fits in the software packages", different fit functions or numerical implementations of the optimization?

---

## Referee Comment (RC2) · Anonymous Referee #3 · 15 Jun 2017

The study by Y. Wang et al. reports on a detailed evaluation of state-of-science retrievals of HONO column densities from MAX-DOAS measurements of scattered UV radiation during an intensive measurement campaign in 2013 in Mainz, Germany. Following a comparison of HONO columns retrieved by 11 different groups, the authors present an in-depth analysis of the retrieval settings required for optimal fitting. This analysis also allows them to present a characterization and breakdown of the error budget of the HONO retrievals. Both aspects of the paper are scientifically important, very suitable for AMT, and in my opinion help to improve and better understand the

MAX-DOAS HONO retrievals.

Strong about this manuscript is that a substantial number of dedicated and relevant sensitivity tests have been carried out to improve the fitting approach, and at the same time characterize the fitting errors. The team makes a strong case that using sequential reference spectra instead of once-per-day noontime reference spectra works best, that water vapour absorption should be accounted for in the fit, and that the 335-373 nm fitting window gives most robust retrieval results. The comparison between the sensitivity study results and the discrepancies between HONO columns observed by different groups provides excellent potential to interpret theoretical and practical uncertainties in the retrievals.

I recommend that the paper is published in AMT, but the authors should first clarify a number of issues listed below, and make the manuscript much better readable.

Major issues

1. The title does not cover the aspect of error analysis that is certainly an important component of this paper. I suggest modifying the title accordingly.

2. The paper is too long. In many places too much information is provided. There are too many references in the text to the Supplementary Material and such interruptions prevent a smooth read. The manuscript should be streamlined in many places. As an example, on page 8, L31-32 and P9, L1-15, much of the text is about supplementary figures supporting the material in Figs. 4 and 5. Isn't the material presented in Figs. 5 and 5 convincing enough to stand on its own? It would be more logical to discuss the results shown in Fig. 4 and 5 more extensively and only at the last instance mention that there is support to be found in the supplementary figures. Another option to make the manuscript more concise is to refrain from giving all of the available information for both the FRS and noontime reference spectrum once the recommendation is given to prefer the FRS method. The same holds for the fitting windows that are ultimately not used.

3. I'm not sure if the order of the sections is optimal. If I'm correct, the 11 retrieval groups use the optimal fitting window (335-373 nm) and settings to obtain their results presented in section 3, but the motivation for this is only given in section 4. Isn't it better to present the sensitivity studies and corresponding recommendations before the actual intercomparison? This would also prevent the need to point forward to sections still to come (e.g. on P6, L14-15 "see Section 4.1")

4. The text in the manuscript is sometimes too vague. For instance in the abstract, the last sentence reads "However, systematic uncertainties limit the reliability of the results." Since you have a pretty decent quantitative estimate of the systematic error of the HONO columns, please indicate what you think is the detection limit, and how frequently you think this is being exceeded in practice. This gives potential users of the data a sense of the usefulness of the HONO retrievals, for instance in the context of the diurnal cycle of HONO columns. Also, see many minor comments below, asking for clarifications.

5. The role of clouds in the retrieval remains under-exposed. It would be interesting to distinguish the quality of the spectral fits under cloudy and clear-sky conditions.

Minor issues

P2, L14: "of the fitted from the simulated real HONO delta SCDs". Hard to follow, please rephrase.

P2, L21: "tropospheric atmosphere" → troposphere

P3, L26: I think it would be appropriate to introduce the 11 groups participating in the MAD-CAT campaign here.

P4, L22: "seven of all of the eleven" → Seven of the eleven . . .

P4, L30: repetitive to mention the 12 June – 5 July period here since it was in 2.1

P5, L15: it is unclear at this stage what sigma^2 NO2 represents and what it is used

for. This has to do with the ordering of the section (were section 3 and 4 reversed at the last minute?)

P6, L29-30: is there any physical or chemical reason why HONO dSCDs are high on 3 July 2013?

P7, L31-32: please clarify what 0.01 means here. How should the number be interpreted?

End of P9, lines 1-3 op P10: difficult to follow. I think section 3.3 is in need of a clear conclusion on what we have learned from the statistical comparison. Instead, we end with a quite detailed, unsatisfying comment on something that could be wrong with one particular instrument.

P10, L5-6: "real atmospheric values for real MAX-DOAS measurements"?

P10, L30: "than the half of that"→ than half of that

P11, L17-18: nonlinear fits . . . were not included

P12, L14: can you elaborate on the increase in HONO with an increase in H2O delta SCDs? Is there a good reason to expect this?

P12, L24: peek→peak

P12, L26-27: this has been said already.

P12, L29: "bands" or are they rather lines?

P14, L27: dependence of the Ring spectrum

P15, L6-7: it would be helpful to quantify here what variations you think are due to different Ring settings. This helps is evaluating the overall error budget of the HONO retrievals.

P16, L7: non-linear

P16, L15-16: same as for section 4.5: please quantify the error associated with the intensity offset uncertainty, and conclude as to its relevance.

P16, L25: "instrumental function"→ instrument transfer function or slit function?

P17, L4: "noises" → noise

P18, L8: please calrify what the correlation coefficients refer to.

P18, L10: it would be useful here to explain the typical diurnal variation in HONO, and make clear that the retrievals are able to capture the temporal changes to large extent. Perhaps also indicate when (what column densities, those typically around noon?) the retrievals are running into detection limit issues.

P18, L15: before the paragraph ends, I think

---

## Author Comment (AC1) · 27 Jul 2017

**Reply to Ref. #1, M. Wenig**

First of all we want to thank this reviewer for the positive assessment of our manuscript and the constructive and helpful suggestions.

General comments
The paper describes a very extensive comparison study of several MAX-DOAS HONO measurements from different groups. Using coinciding measurements in Mainz as well as modeled spectra, their comparison can focus on different aspects of the instrument characteristics and retrieval process. The paper addresses relevant scientific questions since the findings can be used for improving existing retrievals and are very valuable for future instrument and algorithm design. It is more an evaluation of existing methods rather than the introduction of a novel concept, but the methods and conclusions are very solid. At some points the paper is a little vague, when only speculations are provided instead of explanations for observed differences. The title is appropriate and the abstract summarizes the presented work well. The paper is a little long, especially in combination with the supplementary material. The paper should be readable without the supplement as it should only provide extra material if the reader wants to go into more details. Overall the manuscript is of high scientific quality and I recommend publication after some minor corrections and clarifications as listed in the following

Author reply:
Many thanks for the positive assessment! We modified the paper based on the comments from you and reviewer 2. In order to make the main text readable without the supplement, we added some important numbers in the main part of the paper (in the parts related to the supplementary figures). We hope the revised manuscript is more smoothly readable.

**Specific Comments:**
1) L13p2 "uncertainty from different sources" sounds confusing, how about "uncertainty from all the different sources" or "overall uncertainty"?
Author reply: we modified it as "overall systematic uncertainty".

2) L14p2 "the systematic bias of the fitted from the simulated real HONO delta SCDs is" is grammatically challenging as well.
Author reply: the sentence is deleted.

3) L18p2 The sentence "However, systematic uncertainties limit the reliability of the results." is very general. What do you know about systematic errors? If the bias is known, you can correct for it. If not, how reliable are the results?
Author reply: We modified the sentence based on your comment and the comment from the reviewer 1. The new sentence is "In summary for most of the MAX-DOAS instruments for elevation angles below $5°$, half daytime measurements (usually in the morning) can be above the detection limit of the HONO delta SCD of $0.2 \times 10^{15}$ molecules cm$^{-2}$ with an uncertainty of $\sim 0.9 \times 10^{15}$ molecules cm$^{-2}$."

4) L25ffp5 When you define the different quantities, please provide equations. That would make it easier later on in the paper if you could just refer to the variable. For example what is the difference between delta SCD with sequential FRS and dSCD?

Author reply: We add the equation derivations in the revised manuscript to define the dSCD and delta SCDs clearly in section 3.1.

5) L4ffp6 When you describe Fig. 1 you mention the different fit results for O3, NO2 and BrO but for HONO you just write "The HONO absorption structures are well retrieved using both types of FRS.", but the fit parameters are 5.5 and 6.9 10^15 molec. cm^-2, so quite different.

Author reply: We modified the sentence as "The difference of retrieved HONO dSCDs between the two fits is mainly due to the different HONO absorption in the two FRS. The same reason also leads to the differences of retrieved dSCDs of the other trace gases. The difference is substantially larger for the trace gases with considerable stratospheric contributions, e.g. $O_3$ and BrO, because the stratospheric light paths around noon for the daily noon FRS are much shorter than those during sunset or sunrise."

6) L31fp6 You write "because of unknown instrumental problems, CMA and Boulder didn't participate in the comparisons of the delta SCDs for a sequential FRS and dSCDs for a daily noon FRS, respectively", you could mention that the other measurements were not affected.

Author reply: we add the clarification of "but other instruments are not affected."

7) L1fp7 In the sentence "all the instruments capture well the diurnal evolution and elevation angle dependence of the HONO delta SCDs." "capture well" sounds very general, you could be more specific by mentioning if they agree within the uncertainties or if there are significant differences.

Author reply: Since a detailed quantive analysis of the inter-comparisons results is given in section 3.3, here we only mention that in general similar HONO results are observed by the different instruments. In the revised paper we modified the sentence as "from all instruments a similar diurnal evolution and elevation angle dependence of the HONO delta SCDs is retrieved." and clarified that "A detailed quantitative analysis of the deviations of the HONO results between the instruments is provided in the statistical analysis in section 3.3."

8) L10ffp8 You use the average of selected instruments as the reference values, but then you compare all instrument to that reference (l13p8,l11p9), that's not very consistent. How about using a median or something similar for reference instead?

Author reply: Because the Boulder and CMA instruments are affected by some unknown instrumental problems, we prefer to only include the more stable instruments. We clarified this point in the revised manuscript as "In addition the selection is also because the Boulder and CMA instruments are affected by some unknown instrumental problems."

9) L2fp10 Again, an "unknown problem" can always be the reason for discrepancies, can you be more spefic?

Author reply: Sorry, we can't give the specific reason. We only can speculate this is due to an instrumental problem, like spectrometer or motor. We specify the problem as "an unknown instrumental problem" in the revised manuscript.

10) L12p10 You mention "Note that there is no random noise added into the synthetic spectra." But in Sec. 4.9 (l4fp17,also table S1) a SNR of 3000 is mentioned that is added to the synthetic spectra, so why not in Sec. 3.4?

Author reply: The synthetic spectra used here are to quantify the systematic difference of DOAS fits implemented by different groupds and to quantify the systematic difference of retrieved results from the truth. Moreover, in section 4.9 the effect of noise is investigated and it was found that the noise added in the synthetic spectra doesn't lead to any considerable systematic deviation. Therefore it is meaningful to investigate systematic effects on the HONO retrieval based on synthetic spectra without noise. We added a clarification in the revised manuscript as "Note that there is no random noise added into the synthetic spectra due to the objective to quantify the systematic difference of the retrieved values from the truth and the negligible effect of noise on the systematic difference concluded in Section 4.9. ".

11) L29p13 You write "The Thalman O4 cross section at 203k orthogonalised to that at 293k is calculated by the orthogonalisation based on Gram-Schmitt's algorithm and is shown in Fig.9." and Fig. 9's caption says "normalized absorption cross sections". How were they normalized or related to that what scalar product was used for the orthogonalization algorithm, e.g. is a polynomial removal part of the scalar product?

Author reply: We modified the description in the revised manuscript as " The Thalman $O_4$ cross section at 203k is orthogonalised to that at 293k based on Gram-Schmitt's algorithm using a polynomial of two degrees. The orthogonalised $O_4$ cross section is normalized by an arbitrary factor to be shown in a comparable scale with other cross sections in Fig.9."

12) L29p15 The degree of the polynomial was "arbitrarily selected" as five, why not 3? Fewer parameters to fit usually make the optimization more stable.

Author reply: Because of the weak dependence of HONO results on polynomial degrees in 335-373nm, the parameter can be freely selected between 3 to 5. In addition because in some cases short time variations of the sky conditions might happen in real measurements, we selected a higher polynomial degree, which can better account for such changes. We clarified this point in the revised manuscript: "The effect of the degree of the polynomial on the HONO results in the wavelength range 335-373 nm is small. However because in some cases short time variations of the sky conditions might happen in real measurements, we recommend to select a higher polynomial degree, which can better account for such changes. A fifth degree polynomial is used for HONO retrievals in this study. "

13) L12p16 "In spite of these possible interferences, taking into account typical instrumental problems (like spectrograph straylight), the consideration of an intensity offset correction in the fit is still recommended for the HONO retrieval." I agree, but can you underline this statement with some data or estimates? How do you know which influence is more important?

Author reply: We can not quantify the straylight effect. We clarified this point in the revised manuscript: "The effect of spectrograph straylight can not be quantified here because it needs a sophisticated Lab measurement, which was not operated during the campaign.".

14) L23fp18 What do you mean exactly with "systematic differences [. . .] caused by implementation of DOAS fits in the software packages", different fit functions or numerical implementations of the optimization?

Author reply: The exact reason in the codes of software is unknown here. We can only generally attribute the differences of HONO results to the differences of the codes of DOAS software. We

clarified it in the revised manuscript: "However the exact reason in the codes of software, which cause the difference, is unknown here. We can only generally attribute the differences of HONO results to the differences of the codes of DOAS software."

---

## Author Comment (AC2) · 27 Jul 2017

**Reply to Ref. #2**

First of all we want to thank this reviewer for the positive assessment of our manuscript and the constructive and helpful suggestions.

General comments
The study by Y. Wang et al. reports on a detailed evaluation of state-of-science retrievals of HONO column densities from MAX-DOAS measurements of scattered UV radiation during an intensive measurement campaign in 2013 in Mainz, Germany. Following a comparison of HONO columns retrieved by 11 different groups, the authors present an in-depth analysis of the retrieval settings required for optimal fitting. This analysis also allows them to present a characterization and breakdown of the error budget of the HONO retrievals. Both aspects of the paper are scientifically important, very suitable for AMT, and in my opinion help to improve and better understand the MAX-DOAS HONO retrievals.
Strong about this manuscript is that a substantial number of dedicated and relevant sensitivity tests have been carried out to improve the fitting approach, and at the same time characterize the fitting errors. The team makes a strong case that using sequential reference spectra instead of once-per-day noontime reference spectra works best, that water vapour absorption should be accounted for in the fit, and that the 335-373 nm fitting window gives most robust retrieval results. The comparison between the sensitivity study results and the discrepancies between HONO columns observed by different groups provides excellent potential to interpret theoretical and practical uncertainties in the retrievals.
I recommend that the paper is published in AMT, but the authors should first clarify a number of issues listed below, and make the manuscript much better readable.

Author reply:
Many thanks for the positive assessment! We modified the paper based on the comments from you and reviewer 1. In order to make the main text readable without the supplement, we added some important numbers in the main part of the paper (in the parts related to the supplementary figures). We hope the revised manuscript is more smoothly readable.

**Major issues:**

1) The title does not cover the aspect of error analysis that is certainly an important component of this paper. I suggest modifying the title accordingly.
Author reply:
We followed the suggestion and added "error budget" in the title.

2) The paper is too long. In may places too much information is provided. There are too many references in the text to the Supplementary Material and such interruptions prevent a smooth read. The manuscript should be streamlined in many places. As an example, on page 8, L31-32 and P9, L1-15, much of the text is about supplementary figures supporting the material in Figs. 4 and 5. Isn't the material presented in Figs. 5 and 5 convincing enough to stand on its own? It would be more logical to discuss the results shown in Fig. 4 and 5 more extensively and only at the last instance mention that there is support to be found in the supplementary figures. Another option to

make the manuscript more concise is to refrain from giving all of the available information for both the FRS and noontime reference spectrum once the recommendation is given to prefer the FRS method. The same holds for the fitting windows that are ultimately not used.

Author reply:
Thanks for pointing the problem out. Based on the suggestion of the reviewer, in order to make the main text independent on the supplement, we modified the section 3.3 and section 4, in which many relevant figures are given in the supplement. In the revised version the main results are directly described in the main text. The reader doesn't need to see the figures in the supplement. These figures are only referenced in the text to allow the most interested reader to see the detailed results.
Many thanks also for the suggestion of the second option! However, we decided not to follow this suggestion. It is true that the recommended settings, especially the selection of the wavelength range and the selection of the FRS are mainly derived from the sensitivity analyses in section 4. However, it is also important to derive the same conclusion from the comparison of the results from the different instruments, because usually not all apects relevant for real measurements can be covered by the analysis of synthetic spectra. Thus we prefer to keep the retrieval results using different FRS in the main part of the paper.

3) I'm not sure if the order of the sections is optimal. If I'm correct, the 11 retrieval groups use the optimal fitting window (335-373 nm) and settings to obtain their results presented in section 3, but the motivation for this is only given in section 4. Isn't it better to present the sensitivity studies and corresponding recommendations before the actual intercomparison? This would also prevent the need to point forward to sections still to come (e.g. on P6, L14-15 "see Section 4.1")

Author reply:
The suggestion of the reviewer is also logic. However the relationships between the different parts of the paper are not only valid in one direction. For example, also from the experimental results part of the sensitivity studies performed in section 4 are motivated. Thus we prefer to keep the general structure of the paper as it is. However to better guide the reader through the manuscript, we added a clarification in the beginning of section 3:
"HONO presents prominent absorption structures in the spectral range from 335 to 390 nm. The DOAS technique (Platt and Stutz, 2008, and references therein) can be applied to spectra of scattered sunlight to retrieve SCDs of HONO. In this section we present the inter-comparison of HONO SCD results derived from real measurements and synthetic spectra between the participants. For the analyses of both sets of spectra, recommended baseline settings for the DOAS spectral analysis are provided. These baseline setting are derived from the sensitivity studies outlined in section 4 and also based on the experiences in Hendrick et al. (2014). The details of the baseline setting are given in Table 2 and described in section 3.1."
The beginning of section 3.1 is also modified accordingly.

4) The text in the manuscript is sometimes too vague. For instance in the abstract, the last sentence reads "However, systematic uncertainties limit the reliability of the results." Since you have a pretty decent quantitative estimate of the systematic error of the HONO columns, please indicate what you think is the detection limit, and how frequently you think this is being exceeded in practice. This gives potential users of the data a sense of the usefulness of the HONO

retrievals, for instance in the context of the diurnal cycle of HONO columns. Also, see many minor comments below, asking for clarifications.

Author reply:
We modified the sentence in the abstract as "In summary for most of the MAX-DOAS instruments for elevation angle below 5°, half daytime measurements (usually in the morning) can be over the detection limit of HONO delta SCD of $0.2\times10^{15}$ molecules $cm^{-2}$ with a uncertainty of ~$0.9\times10^{15}$ molecules $cm^{-2}$."

5) The role of clouds in the retrieval remains under-exposed. It would be interesting to distinguish the quality of the spectral fits under cloudy and clear-sky conditions.

Author reply:
Thanks for the suggestion. We checked the fit error of HONO dSCD under cloudy and clear days. We found the errors are quite similar. The reason is the most of instruments can automatically change the exposure time of spectrometer based on the sunlight intensity. Therefore the similar exposure saturation level is reached during clear and cloudy days. We clarified this point for the discussion of Fig. 4 in section 3.2 of the manuscript: "In addition fit errors of HONO dSCD under cloudy and clear days are quite similar due to the fact that the MAX-DOAS instruments automatically change the exposure time of spectrometer based on the brightness of the sky. Therefore the similar exposure saturation level is reached during clear and cloudy days.".

**Minor issues:**
1) P2, L14: "of the fitted from the simulated real HONO delta SCDs". Hard to follow, please rephrase.
Author reply: The sentence is deleted in the revised version.

2) P2, L21: "tropospheric atmosphere" → troposphere
Author reply: corrected.

3) P3, L26: I think it would be appropriate to introduce the 11 groups participating in the MAD-CAT campaign here.
Author reply: considering that not all the groups joined this study, we add the MAD-CAT website link in which the 11 groups are listed.

4) P4, L22: "seven of all of the eleven" → Seven of the eleven.
Author reply: corrected.

5) P4, L30: repetitive to mention the 12 June – 5 July period here since it was in 2.1
Author reply: The sentence is deleted.

6) P5, L15: it is unclear at this stage what sigma^2 NO2 represents and what it is used for. This has to do with the ordering of the section (were section 3 and 4 reversed at the last minute?)
Author reply: As the reply to the major issue 3, we prefer to keep the structure as it is. For this point, we added a note of "(the details are given in section 4.5)".

7) P6, L29-30: is there any physical or chemical reason why HONO dSCDs are high on 3 July 2013

Author reply: The high HONO in the morning is not only on 3 July 2013, but also on many other days (see Fig. 2a). Photolysis and high $NO_2$ concentration can cause the substantial high HONO concentration in the morning. We clarified this point in the revised manuscript: "The large HONO values in the morning could be due to a high $NO_2$ concentration ($NO_2$ dSCD of up to $1\times17$ molecules cm$^{-2}$) and a fast photolysis of HONO (e.g. Hendrick et al., 2014).". Nevertheless, the chemical sources of HONO is not the topic of this study, therefore we don't discuss this point deeply.

8) P7, L31-32: please clarify what 0.01 means here. How should the number be interpreted?

Author reply: It is clarified as "1% of the mean intensity in the fit window" in the revised version.

9) End of P9, lines 1-3 op P10: difficult to follow. I think section 3.3 is in need of a clear conclusion on what we have learned from the statistical comparison. Instead, we end with a quite detailed, unsatisfying comment on something that could be wrong with one particular instrument.

Author reply: we added the general conclusion in the end of section 3.3 in the revised manuscript as "In general the consistent temporal variation and elevation angle dependence of the HONO delta SCDs and dSCDs have been retrieved from the different instruments. The discrepancy of HONO dSCDs from the fits with a daily noon FRS between the instruments is systematically larger than that of HONO delta SCDs, which can be consistently retrieved from the fits with a daily noon and a sequential FRS."

10) P10, L5-6: "real atmospheric values for real MAX-DOAS measurements"?

Author reply: The sentence is modified as "In general it is difficult to quantify the biases of the retrieved HONO dSCDs with respect to the reality in the atmosphere for real MAX-DOAS measurements as the true HONO column is not known."

11) P10, L30: "than the half of that"→ than half of that

Author reply: corrected.

12) P11, L17-18: nonlinear fits . . . were not included

Author reply: corrected.

13) P12, L14: can you elaborate on the increase in HONO with an increase in H2O delta SCDs? Is there a good reason to expect this?

Author reply: The correlation is probably due to the interference of the absorption structures of $H_2O$ with those of HONO in the DOAS fits. As demonstrated in the paper, if the $H_2O$ cross section is not included in the DOAS fit, the $H_2O$ absorption can contribute to residual structures, which can interfere with the retrieved HONO structures. The interference is stronger for larger $H_2O$ absorptions. We modified the sentence to make the point clear in the revised manuscript: "These findings demonstrate that the $H_2O$ absorption could mainly contribute to the residual structure around 363nm if the $H_2O$ cross section is not included in the DOAS fit, and can considerably interfere with the HONO absorption. And the interference is stronger for larger $H_2O$ absorptions.".

14) P12, L24: peek→peak

Author reply: corrected

15) P12, L26-27: this has been said already.
Author reply: The scaling factor of $H_2O$ cross section is only mentioned in section 3.4 for the RTM simulation of synthetic spectra. Here the factor is mentioned again for the DOAS fit. They are for different activity. Therefore we prefer to keep it.

16) P12, L29: "bands" or are they rather lines?
Author reply: The HONO absorption structures are smooth as shown in Fig. 9. Therefore "bands" are better than "lines".

17) P14, L27: dependence of the Ring spectrum
Author reply: corrected

18) P15, L6-7: it would be helpful to quantify here what variations you think are due to different Ring settings. This helps is evaluating the overall error budget of the HONO retrievals.
Author reply: We added the numbers as "(about 0.35, 0.2, and 0.12 $\times 10^{15}$ molecules cm$^{-2}$ in the spectral ranges of 335-361, 335-373, and 335-390 nm, respectively on averaged)" in the revised manuscript.

19) P16, L7: non-linear
Author reply: corrected

20) P16, L15-16: same as for section 4.5: please quantify the error associated with the intensity offset uncertainty, and conclude as to its relevance.
Author reply: The numbers are added as "which is 0.55, 0.35, and 0.25 $\times 10^{15}$ molecules cm$^{-2}$ in the spectral ranges of 335-361, 335-373, and 335-390 nm, respectively on averaged" in the revised manuscript.

21) P16, L25: "instrumental function" $\rightarrow$ instrument transfer function or slit function?
Author reply: slit function. We modified all the word of "instrumental function" as "slit function" in the revised manuscript.

22) P17, L4: "noises" $\rightarrow$ noise
Author reply: corrected

23) P18, L8: please calrify what the correlation coefficients refer to.
Author reply: we clarified it as "The correlation coefficients of HONO delta SCDs derived from the different instruments with the reference values" in the revised manuscript.

24) P18, L10: it would be useful here to explain the typical diurnal variation in HONO, and make clear that the retrievals are able to capture the temporal changes to large extent. Perhaps also indicate when (what column densities, those typically around noon?) the retrievals are running into detection limit issues.
Author reply: We added the description as "The maximum value of the HONO delta SCDs is about $6 \times 10^{15}$ molecules cm$^{-2}$ and usually in the morning. The HONO delta SCD rapidly decrease

after sunrise due to the photolysis of HONO, and below the detection limit of $0.2 \times 10^{15}$ molecules cm$^{-2}$ in the afternoon." in the revised manuscript.

25) P18, L15: before the paragraph ends, I think

Author reply: We moved the whole sentence of "In addition the deviations of the HONO dSCDs derived from the fits with daily noon FRS between the instruments are generally larger than those of the HONO delta SCDs mainly due to the different HONO absorptions in the noon FRS and the interferences by the stratospheric species, e.g. ozone." before the sentence of "Furthermore, there are no considerable systematic differences of the HONO delta SCDs from the fits with the sequential FRS and the daily noon FRS for all the instruments except the mini MAX-DOAS instrument." in the revised manuscript.